# Brain Tumors, AI and Psychiatry: Predicting Tumor-Associated Psychiatric Syndromes with Machine Learning and Biomarkers

**DOI:** 10.3390/ijms26178114

**Published:** 2025-08-22

**Authors:** Matei Șerban, Corneliu Toader, Răzvan-Adrian Covache-Busuioc

**Affiliations:** 1Puls Med Association, 051885 Bucharest, Romania; mateiserban@innbn.com (M.Ș.); razvancovache@innbn.com (R.-A.C.-B.); 2Department of Neurosurgery, “Carol Davila” University of Medicine and Pharmacy, 050474 Bucharest, Romania; 3Department of Vascular Neurosurgery, National Institute of Neurology and Neurovascular Diseases, 077160 Bucharest, Romania

**Keywords:** brain tumor psychiatry, neuro-oncology, IDH-mutant glioma, 2-hydroxyglutarate (2-HG), neuroimmune dysregulation, glutamatergic dysfunction, epigenetic reprogramming, lesion-network disruption, paraneoplastic psychosis, AI-driven neuropsychiatric diagnostics

## Abstract

Brain tumors elicit complex neuropsychiatric disturbances that frequently occur prior to radiological detection and hinder differentiation from major psychiatric disorders. These syndromes stem from tumor-dependent metabolic reprogramming, neuroimmune activation, neurotransmitter dysregulation, and large-scale circuit disruption. Dinucleotide hypermethylation (e.g., IDH-mutant gliomas), through the accumulation of 2-hydroxyglutarate (2-HG), execute broad DNA and histone hypermethylation, hypermethylating serotonergic and glutamatergic pathways, and contributing to a treatment-resistant cognitive-affective syndrome. High-grade gliomas promote glutamate excitotoxicity via system Xc^−^ transporter upregulation that contributes to cognitive and affective instability. Cytokine cascades induced by tumors (e.g., IL-6, TNF-α, IFN-γ) lead to the breakdown of the blood–brain barrier (BBB), which is thought to amplify neuroinflammatory processes similar to those seen in schizophrenia spectrum disorders and autoimmune encephalopathies. Frontal gliomas present with apathy and disinhibition, and temporal tumors lead to hallucinations, emotional lability, and episodic memory dysfunction. Tumor-associated neuropsychiatric dysfunction, despite increasing recognition, is underdiagnosed and commonly misdiagnosed. This paper seeks to consolidate the mechanistic understanding of these syndromes, drawing on perspectives from neuroimaging, molecular oncology, neuroimmunology, and computational psychiatry. Novel approaches, including lesion-network mapping, exosomal biomarkers or AI-based predictive modeling, have projected early detection and precision-targeted interventions. In the context of the limitations of conventional psychotropic treatments, mechanistically informed therapies, including neuromodulation, neuroimmune-based interventions, and metabolic reprogramming, are essential to improving psychiatric and oncological outcomes. Paraneoplastic neuropsychiatric syndromes are not due to a secondary effect, rather, they are manifestations integral to the biology of a tumor, so they require a new paradigm in both diagnosis and treatment. And defining their molecular and circuit-level underpinnings will propel the next frontier of precision psychiatry in neuro-oncology, cementing the understanding that psychiatric dysfunction is a core influencer of survival, resilience, and quality of life.

## 1. Introduction

### 1.1. Background and Rationale

Neuro-oncology meets psychiatry: a watershed area of research, defined by the insidious and pervasive psychiatric effects of cerebral injury in the context of malignant disease. Traditionally, brain tumor-induced psychiatric symptoms were demoted to a secondary status, being seen either as reactive psychological distress or misdiagnosed as the psychosocial burden accompanying malignancy [1]. However, new evidence points to these symptoms not as incidental, but rather as an integral part of tumor pathophysiology, through structural, metabolic, neurochemical, and immune-mediated mechanisms. The phenomenon described above has important clinical implications due to the psychiatric presentations often being a precursor to tumor discovery, leading to delays in appropriate oncological treatment and consistently impacting treatment compliance and prognosis [2]. The neuropsychiatric disturbances associated with brain tumors are often underdiagnosed and poorly characterized, and their pathophysiological basis, diagnostic approaches, and management strategies deserve re-evaluation [3].

The central nervous system (CNS) is an important focus for psychiatric symptoms in brain tumors, where the presenting symptoms often span a broad clinical spectrum from subtle cognitive and personality change to severe affective dysregulation or psychosis and catatonia. Epidemiological studies show that 30–70% of patients with primary or metastatic brain tumors experience clinically significant psychiatric symptoms at some point during the course of their disease [4]. Neuropsychiatric symptoms may occur early in gliomas, especially high-grade variants such as glioblastoma multiforme (GBM), with depression and cognitive decline often preceding those changes reported by radiological imaging. By contrast, low-grade gliomas classically present with insidious personality changes and executive dysfunction, leading to misdiagnosis as primary psychiatric illnesses [5].

Secondary brain lesions, especially from lung, breast, and melanoma primaries, add further complicating layers, often psychiatric syndromes related to multifocal involvement, paraneoplastic process and systemic metabolic dysregulation [6].

Owing to the functional specialization of cortical and subcortical circuits controlling mood, cognition, and behavior, the neuroanatomical localization of tumors is a key determinant of the contained psychiatric symptomatology. Disruption of executive function, impulse control, and emotional regulation for frontal lobe tumors leads to apathy, disinhibition, and depressive states [7]. Involvement of the temporal lobe is related to emotional lability, psychotic symptoms and episodic memory deficits, often presenting as schizophrenia spectrum disorder or psychosomatic syndromes related to temporal lobe epilepsy. Tumors arising in the parietal and occipital lobes, though less commonly associated with primary psychiatric syndromes, have also been associated with perceptual disturbances, delusional misidentification syndromes, and deficits in visuospatial cognition [8]. Subcortical neoplasms involving the basal ganglia and limbic system also add to neuropsychiatric dysregulation by dopaminergic disturbance considered to be manifested as anhedonia, psychomotor slowing, and affective blunting [9].

Beyond structural effects, tumor-driven neuroinflammation and immune dysregulation are increasingly recognized as contributors to psychiatric symptoms. Gliomas generate an immuno-suppressive milieu, with distorted cytokine signaling (IL-6, TNF-α, IFN-γ) and microglial activation that disrupts neuronal function [10]. Another emerging mechanism involves tumor-derived exosomal signaling. Glioma extracellular vesicles (EVs), enriched with microRNAs such as miR-21 and miR-155, can influence synaptic plasticity and neurotransmission, contributing to depressive and cognitive symptoms [11]. In addition, blood–brain barrier compromise in gliomas allows peripheral immune cell infiltration, amplifying neuroinflammatory cascades similar to those seen in autoantibody-mediated psychiatric disorders, such as anti-NMDA receptor encephalitis [12].

Another emerging area of investigation is tumor metabolism and its putative role in psychiatric symptomatology. High-grade gliomas also modulate glutamate homeostasis through upregulation of the system xc- transporter, resulting in supraphysiological glutamate release, excitotoxicity, and seizure susceptibility—an effect that may also contribute to the high rates of mood instability and cognitive impairment seen in these patients [13]. Changes in the tryptophan-kynurenine pathway, a known metabolic axis in neuroinflammation and psychiatric disorders, have also been associated with glioma-associated depressive symptoms, as elevated kynurenine end products can disturb serotoninergic signaling [14]. In addition, the disruption of dopaminergic and cholinergic pathways—the involvement of the basal ganglia or systemic paraneoplastic processes—also contribute to cognitive and affective impairment in metastatic brain tumors, emphasizing that psychiatric disturbances in the cancer patient cannot be attributed to simple mechanisms, but emerge from a complex interplay of neurochemical, inflammatory, and metabolic dysregulations [15].

Although these pathophysiological links are becoming increasingly recognized, the distinction between tumor-induced psychiatric syndromes and primary psychiatric disorders continues to pose substantial diagnostic difficulties. In fact, a substantial number of patients found to have undiagnosed brain tumors present with isolated psychiatric symptoms at first and are prescribed psychotropic medications for months until neuroimaging reveals an underlying neoplasm [16]. This delayed diagnosis has a significant impact, while survival rates and outcomes are markedly enhanced by early tumor detection. In addition, untreated psychiatric disorders decrease adherence to treatment, compromise patients’ autonomy in medical decision-making, and adversely affect the overall functional prognosis. The need to address these neuropsychiatric manifestations is certainly not an ancillary to oncological care, but an integral part of holistic patient management [17].

### 1.2. Objectives and Scope

Hence, in order to advance the understanding of psychiatric manifestations associated with cerebral tumors, an integrative approach requiring both neuro-oncology, neuropsychiatry and molecular neuroscience is necessary. Psychiatric manifestations in brain tumor patients result from synergistic effects of tumor biology, neuroinflammation and altered neurotransmission, but their mechanistic basis has not yet been fully specified. This work aims to integrate emerging evidence that tumor-induced perturbations in neural circuitry, metabolic pathways, and immune signaling manifest as unique neuropsychiatric syndromes. Identification of these mechanistic drivers may allow for improved diagnostic accuracy and focused therapeutic development.

One of the key areas is enhancing early recognition of tumor-induced psychiatric syndromes, since diagnostic delays continue to be a major impediment to timely treatment. Emerging data from functional neuroimaging, radiomics-based modeling, and metabolic PET tracers reveal coherent patterns of altered neurophysiology in tumor-related psychiatric syndromes that may help distinguish them from primary psychiatric disorders. Concurrent advances in liquid biopsy techniques, such as circulating tumor DNA, glioma specific exosomal microRNAs and paraneoplastic autoantibodies, are allowing for the increasingly less invasive diagnostic methods that could be of utility in neuro-oncological screening in psychiatric patients. Assessing the clinical utility of these approaches will be the challenge to incorporating them into routine clinical practice.

Neuroimmune modulation, disparate BBB permeability, and psychotropics-oncologic medication interactions make optimizing psychiatric management in patients with brain tumors especially challenging. Pharmacogenomic advances, precision psychiatry, as well as neuromodulatory approaches including TMS and DBS are being integrated into potential means to tailor psychiatric care to this cohort. It is also important to understand how oncological treatments—particularly surgical resection, radiotherapy, and chemotherapy—influence neuropsychiatric trajectories, given their long-term effects on cognition, mood, and quality of life.

Therefore, improving outcomes requires not only better experiences of interdisciplinary collaborative care, but also evidence-based sharing practices on standardized neuropsychiatric screening protocols. This integration of ML models with multimodal clinical and imaging data may open new doors for predictive diagnostics and personalized treatment approaches. By condensing these advances, this piece seeks to unite core aspects of hypothesis-driving discoveries with the translation process of clinical intervention, allowing for psychiatric symptoms associated with brain tumors to be acknowledged as intrinsic components of disease pathology and proactively treated to an equivalent level of resolution as standard oncologic comorbidities.

## 2. Overview of Cerebral Tumors and Pathophysiological Mechanisms

### 2.1. Classification and Epidemiology

Molecular neuropathology has delineated a new reality of the classification of cerebral tumors, whereby a paradigm has shifted from histological-based tumors to genomically defined subtypes. The WHO CNS tumors classification of 2024 is better defined with the incorporation of molecular markers, epigenetic alterations, and transcriptomic profiling which have direct implications for prognosis, treatment response, and psychiatric symptomatology [18,19]. Brain tumors are mostly grouped into gliomas, meningiomas, pituitary adenomas, or embryonal tumors (i.e., childhood brain tumors), while secondary (metastatic) brain tumors originate from systemic malignancies (lung, breast, melanoma, renal cell carcinoma) [20].

Over the years, gliomas have been the most commonly diagnosed form of primary malignant brain tumors with GBM accounting for nearly 50% of all malignant gliomas. Recent meta-analyses and a comprehensive review recently led to the classification of two forms of gliomas according to the mutation status of IDH, highlighting that IDH-mutant gliomas exhibit a slower progression than IDH-wildtype gliomas which are more invasive with aggressive invasion and widespread peritumoral inflammation, contributing to cognitive and affective dysfunction early in the course of these tumors [21]. Importantly, discovery of TERT promoter mutations and EGFR amplifications in high-grade gliomas have confirmed their involvement in tumor-mediated neurotoxicity, neurotransmitter dysregulation, and synaptic remodeling that could serve to enhance psychiatric symptoms [22].

Meningiomas are commonly benign lesions but are now acknowledged as a common cause of location-dependent psychiatric sequelae, especially when involving the parasagittal, sphenoidal, and orbitofrontal regions. It is now understood that tumors can cause symptoms, including apathetic depression, executive dysfunction, and behavioral disinhibition, due to disruption of prefrontal–subcortical networks locally, not through mass effect alone [23]. Prior studies suggest that NF2-mutated meningiomas have a greater burden of peritumoral edema and neuroinflammation, potential contributing factors for neuropsychiatric disturbances even in patients with small, apparently indolent tumors [24].

Pituitary adenomas are composed of secretory cells, represent 15% of all intracranial neoplasms and can have a significant impact on psychiatric health by mass effect on hypothalamic and limbic circuits as well as through endocrine disruption [25]. The side effects of hyperprolactinemia, Cushing’s disease, and acromegaly on affective, cognitive, and psychotic symptoms are well known, with emerging evidence suggesting that dopaminergic dysregulation in prolactinomas produces states of anhedonia and apathy, whereas chronic hypercortisolism induces hippocampal atrophy and depressive phenotypes through glucocorticoid receptor-mediated neurotoxicity [26].

Brain metastases, which occur at a ten-to-one rate compared to primary brain tumors, take on unique neuropsychiatric characteristics due to their diffuse nature, high peritumoral edema, and systemic inflammatory load. Metastatic tumors exhibit particular neuropsychiatric profiles that reflect their primary site of origin (e.g., lung cancer metastases frequently involve the parietal and occipital lobes, leading to visual hallucinations and spatial disorientation; however, melanoma metastases show a predilection for the limbic system, predisposing patients to significant affective instability and agitation) [27]. In addition, HER2-positive breast cancer brain metastases are linked to cognitive dysfunction via neuroimmune changes compounded by BBB disruption, indicating a need for personalized psychiatric intervention strategies that accounts for therapy during metastatic disease [28].

Globally, the epidemiology of CNS neoplasms has changed beyond tumor type due to increased longevity, better systemic cancer control programs, and better neuroimaging. The elderly population with the highest incidence of glioblastomas; however, pediatric brain tumors, including medulloblastomas and diffuse midline gliomas (H3K27M-mutated) show aggressive neurodevelopmental effects resulting in cognitive and emotional maturation [29]. Neuroepidemiological studies have also revealed regional differences in the epidemiology of brain tumors, such as the greater incidence of glioma subtypes in Western countries compared to Asian facial populations and a higher propensity for meningiomas and pituitary adenomas in Asian and African populations, which may represent the role of genetic, environmental, and healthcare access factors [30].

Understanding these epidemiological and molecular features is necessary to accurately relate tumor type and location to neuropsychiatric manifestations and is the foundation of optimizing diagnosis and treatment. Now, next-generation sequencing, epigenetic profiling, and radiomic phenotyping are disrupting the classification and understanding of these tumors, paving the way for precision medicine approaches in neuro-oncology and neuropsychiatry [31,32].

### 2.2. Neuroanatomy and Tumor Location

The psychiatric sequelae of cerebral tumors are heavily contingent on their anatomical location, with dysregulation of large-scale functional networks, white matter connectivity, and neurotransmitter homeostasis resulting in distinct neuropsychiatric syndromes. The traditional approach was to associate particular psychiatric symptoms with tumor location. However, network neuroscience and new neuroimaging methods now demonstrate that tumors have widespread effects on brain-wide function by means of remote diaschisis, compensation through hyperconnectivity and global neural network dysregulation [33].

Although direct invasion of a cortical region can cause focal neuropsychiatric symptoms, recent studies using resting-state functional MRI (rs-fMRI), diffusion tensor imaging (DTI), and tractography-based disconnectome mapping show that many psychophysiological symptoms originate from broader network effects. Tumor-induced disconnection of white matter tracts, destabilization of network architecture, and neurotransmitter imbalances can extend far beyond the site of the main lesion [34].

The frontal lobes are central to executive function, emotional control, and social cognition. Tumors in this region often lead to disinhibition, impulsivity, apathy, and deficits in higher-order reasoning. Involvement of the orbitofrontal cortex (OFC), which regulates affect and reward processing, may result in mood instability, compulsive behaviors, and impaired decision-making. Lesions here have been associated with orbitofrontal syndrome—a pattern of euphoria, hypersexuality, emotional dysregulation, and impulsivity that can mimic mania or frontotemporal dementia [35].

By contrast, dorsolateral prefrontal cortex (DLPFC) tumors impair working memory, cognitive flexibility, and planning. This produces a syndrome of apathetic depression, psychomotor slowing, and executive dysfunction. Hypofrontal metabolism in these patients correlates with reduced meso-cortical dopamine signaling on PET imaging—a pattern also seen in Parkinson’s disease-related cognitive dysfunction [36,37]. Tumors involving the medial prefrontal cortex, including the anterior cingulate cortex (ACC), can cause severe abulia or akinetic mutism. These are disruptions of motivation circuits rather than primary motor impairments. Such syndromes worsen when prefrontal–subcortical white matter tracts are also damaged [38]. Tractography has shown that lesions involving the anterior limb of the internal capsule, superior longitudinal fasciculus, and cingulum bundle interrupt connectivity between the prefrontal cortex, basal ganglia, and limbic system. This can manifest as executive dysfunction, emotional blunting, and diminished insight [39].

The temporal lobes—especially the medial temporal structures—are essential for episodic memory, emotion, and language processing. Tumors in this area often cause affective lability, hallucinations, delusions, and personality changes. Such presentations may resemble schizophrenia spectrum disorders or the psychiatric syndromes seen in temporal lobe epilepsy [40]. Damage to the hippocampus and parahippocampal areas produces anterograde amnesia and confabulation. Lesions of the amygdala and anterior temporal cortex are linked to heightened anxiety, irritability, and paranoia [41]. Involvement of the superior temporal gyrus, which is central to auditory processing, can lead to auditory hallucinations and misperceptions—especially when the lesion affects the dominant hemisphere and its language networks [42].

Tractographic disconnectome studies show that especially temporal tumors disconnect limbic–prefrontal pathways (most notably the uncinate fasciculus and inferior longitudinal fasciculus), which are believed to mediate emotion regulation and to enable the retrieval of memories. These connections are thought to be damaged in Capgras syndrome (delusional misidentification) and Fregoli syndrome (the belief that different people are actually the same person in disguise), among other reality-monitoring deficits [43]. Moreover, secondary epileptiform activity commonly occurs with gliomas and metastatic tumors of the temporal lobes, which provides a basis of paroxysmal mood changes, ictal fear, experiences of déjà vu, and postictal psychosis, complicating the psychiatric picture [44].

Lesions in the parietal lobe disturb body schema, visuospatial processing, and attentional regulation, and have been associated with various forms of anosognosia, hemispatial neglect, and self-perception distortions. Somatoparaphrenia (delusions that a person’s own limb is perceived as belonging to someone else) and asomatognosia (denial of ownership of body) has been associated with right parietal tumors (in the posterior parietal cortex or angular gyrus) [45]. In the left hemisphere, lesions that affect the inferior parietal lobule, supramarginal gyrus, and angular gyrus result in language-related deficits, alexithymia, and impaired emotion recognition, underscoring the role of parietal networks in social cognition and theory of mind. Newer data have also implicated parietal tumors in the disruption of default mode network (DMN) with connectivity predisposing patients to depersonalization, derealization, and cognitive fog [46].

Tumors located in the occipital lobe are classically known to give rise to visual field deficits, but their psychiatric manifestations are becoming ever more widely recognized. Lesions in the lingual gyrus and primary visual cortex can lead to elementary visual hallucinations (flashes, forms, geometric shapes), while damage to the ventral occipitotemporal cortex breaks down higher-order visual processing and can cause visual stimuli to be misinterpreted emotionally [47]. Dysfunction in occipital–limbic networks has been implicated in disorders of visual–limbic integration, such as affective disturbances related to misinterpreted facial expressions and environmental signals, in which occipital–limbic interactions may have a critical role in emotional processing of visual input [48,49].

Thalamic tumors disrupt sensory gating, attention regulation, and arousal, frequently resulting in confusional states, amotivational syndromes, and altered levels of consciousness. Thalamic hypometabolism observed based upon PET imaging studies correlates with cognitive slowing, disorganized thought patterns, and decreased emotional responsiveness, symptoms prominent in both thalamic stroke syndromes as well as in many end-stage neurodegenerative disease states [50]. Lesions of the basal ganglia, particularly those involving the striatum and pallidum, perturb dopaminergic and GABAergic pathways and yield psychomotor slowing, anhedonia, and tumors with Parkinsonian-like features located in the brainstem, particularly the periaqueductal gray and pontine tegmentum, can give rise to catatonic syndromes and autonomic instability as well as dissociative-like states, mirroring their role in emotion–motor integration and autonomic regulation [51].

Newly available AI-based functional network analyses and machine learning-based fMRI models could provide predictive capabilities regarding the effect of tumor-induced network dysfunction on future psychiatric symptom severity, representing a new frontier for precision psychiatry in the context of neuro-oncology. To this end, personalized neuromodulation strategies (e.g., TMS and DBS) are under investigation as potential therapeutic options for tumor-induced neuropsychiatric symptoms due to disruption of neural networks [52,53].

By mapping psychiatric syndromes to tumor location, it may guide and target screening and early treatment. A person presenting with frontal-lobe and parietal-lobe involvement could benefit from initial and repeated executive function testing. Patients with temporal lobe tumors would need structured psychosis and memory screening. The lesion-network information can also provide useful guidance for timing psychiatric referrals, planning rehabilitation and/or family caregiver counseling to prevent issues of acute symptom aggravation, and possibly through adherence to oncologic treatment [54].

In this paper, we intend to show how approaches integrating advanced neuroimaging, network-based modeling, and molecular profiling are advancing our understanding of tumor-related neuropsychiatric syndromes beyond the traditional focal lesion models to inform targeted interventions that address large-scale network disruptions rather than focally based cortical processes. Such a paradigm shift towards connectomic and systems-level approaches constitutes a paradigm shift in neuro-oncology and neuropsychiatry with far-reaching consequences for diagnostics, therapeutic approaches, and future investigations of brain tumor-associated psychotopes [32].

### 2.3. Pathophysiological Mechanisms

Psychiatric sequelae of cerebral tumors result from the interplay between brain invasion, systemic metabolic disturbances and immune dysregulation, and neurobiological derangements that are apparent in regions remote from the tumor site. In addition to mass effect and neuronal compression causing focal deficits, this recent work emphasizes the contribution of novel tumor-specific mechanisms such as tumor-associated neuroinflammation, metabolic reprogramming, epigenetic changes, and dysregulation of neurotransmitters as important determinants of psychiatric outcome [55]. To offer a concise visual overview of these interconnected processes, Figure 1 aims to illustrate in a simplified manner how these mechanisms converge on synaptic dysfunction, maladaptive plasticity, and hyperexcitability, ultimately contributing to psychiatric symptoms. The figure does not attempt to capture every molecular detail but intends to support the reader in grasping the overall conceptual framework discussed in this section. It is these processes that themselves drive maladaptive synaptic plasticity, peritumoral hyperexcitability, dysfunctional large-scale networks, and neurochemical changes that drive neuropsychiatric symptoms. The shift from the lesion-based model towards a network-driven, immuno-metabolic perspective constitutes a paradigm shift in the understanding of the neurobiological underpinnings of brain tumor-associated psychiatric disorders [56].

This schematic intends to illustrate how brain tumors initiate interconnected biological pathways—neuroinflammation (mediated by cytokines such as IL-6, TNF-α, and IFN-γ), metabolic reprogramming (including the Warburg effect and altered glutamine metabolism), epigenetic changes (e.g., DNA methylation), and neurotransmitter dysregulation (involving GABA and serotonin). These processes converge to cause synaptic dysfunction, leading to maladaptive plasticity and hyperexcitability of neural circuits. Together, these alterations contribute to the emergence of diverse psychiatric symptoms in affected patients.

It is important to note that malignant tumors have direct effects on the brain, such as compression, peritumoral edema, and neural network disruption, changing both local and large-scale brain connectivity. Disruption of the cortico-striatal-thalamic loop that modulates mood leads to mood disorder and cognitive slowing, whereas tumors impinging on the cingulum bundle and corpus callosum disrupt interhemispheric cooperation, resulting in alexithymia, apathy, and executive dysfunction [57].

In addition to structural displacement, tumors actively rewire peritumoral circuits, fostering abnormal neuronal hyperactivity and maladaptive network plasticity. Excitotoxicity is promoted by the tumor microenvironment in chronic peritumoral hyperexcitability state of the brain through excessive glutamate release, especially in gliomas, which upregulate the system xc- cystine/glutamate antiporter [58]. In addition to directly dissociating these focal seizures, this aberrant neuronal activity in the absence of overt seizures contributes to psychiatric instability, emotional reactivity, and psychotic episodes that can be evanescent. Subclinical epileptiform activity may persist for long periods of time in limbic and association cortices in slow-growing tumors (e.g., low-grade gliomas), explaining mood disturbances, dissociation, and cognitive fatigue long before obvious neurological symptoms arise [59].

A subset of patients with brain tumors develops paraneoplastic limbic encephalitis, a disorder initiated by onconeural autoantibodies that bind to neuronal receptors and synaptic proteins. Anti-NMDA receptor encephalitis, associated with ovarian teratomas and small-cell lung cancer, presents as agitation, paranoia, auditory hallucinations, and fluctuating cognitive impairment, whereas autoantibodies associated with anti-Hu and anti-Ma2 in small-cell lung carcinoma and testicular germ cell tumors are characterized by depression, amnesia, and rigidity in behavior. These syndromes exemplify how systemic malignancies can provoke immune-mediated neuropsychiatric dysfunction, in the absence of direct CNS invasion [60].

This only enhances psychiatric dysfunction due to tumor-associated metabolic reprogramming. Aerobic glycolysis (the Warburg effect) in turn leads to lactate accumulation, altered glutamine metabolism and disturbed neuronal bioenergetics in many types of brain tumors. Such metabolic dysregulation leads to decreased neurotransmitter production, cognitive slowing, and anhedonia, traits reminiscent of patients with specific mitochondrial diseases and neurodegeneration. Tumor-induced metabolic changes additionally disrupt GABAergic and serotonergic signaling, making individuals vulnerable to irritability, fatigability, and lability of mood and affect [61].

Neurovascular integrity has gained recognition as an important determinant of psychiatric outcome among patients with brain tumors. Tumors interfere with BBB integrity, permitting peripheral cytokines, autoantibodies, and inflammatory mediators to access the CNS, augmenting neuroimmune activation. Through vascular endothelial growth factor (VEGF) secretion, the tumor microenvironment induces aberrant angiogenesis, resulting in leaky, dysfunctional capillaries that compound cerebral edema and impair regional perfusion [62]. In fact, gliomas and brain metastases also inhibit the glymphatic system, which decreases perivascular clearance of neurotoxic proteins and inflammatory debris, also significantly increasing moderating cognitive fog, depressive symptoms, and chronic neuroinflammation in these patients [63].

Neuroinflammation has now become recognized as a major contributor to psychiatric sequelae in glioma patients. Tumors lead to a persistent activation of microglia and astrocytes in the brain, thereby releasing cytokines, such as IL-6, TNF-α, and IFN-γ, which also negatively affect hippocampal neurogenesis and synaptic physiology and ultimately drive depressive-like behaviors. Tumor-derived exosomal microRNAs (miR-21, miR-155) then act on neuronal gene expression, alter glutamatergic homeostasis, and enhance peritumoral excitability, from correlative studies linking glioblastomas to progressive psychiatric deterioration [64].

In addition, tumor cell epigenetic changes modulate neuropsychiatric phenotypes. The IDH-mutant gliomas with their DNA hypermethylation may further alter serotonergic and dopaminergic signaling resulting in partial protection from mood dysregulation. In contrast, IDH-wildtype glioblastomas characterized by hypomethylated, proliferative metabolic profiles are associated with a more aggressive psychiatric and cognitive decline. Compared to conventional approaches, epigenetics-based therapy has provided each clinical diagnosis a unique treatment option which means that manipulation of histone and DNA methylation could represent valuable new treatments for tumor-associated neuropsychiatric dysfunction [65]. Table 1 intends to summarize landmark, original research and clinical trials that have elucidated the complex interplay between tumor biology and neuropsychiatric symptoms. It aims to offer a systematic summary of tumor-specific neuropsychiatric phenotypes, potential pathophysiological pathways, salient clinical features, and translational ramifications for precision psychiatry in neuro-oncology.

Advanced neuroimmune and metabolic agents alongside advanced machine and deep learning neuroimaging are changing the face of psychiatric management in neuro-oncology. ICIs, including anti-PD-1 and anti-CTLA-4 therapies, have transformed cancer treatment but are associated with immune-related neuropsychiatric toxicities, such as encephalitis, mania, and cognitive dysfunction [97]. This understanding will be critical for developing immune-modulating psychiatric interventions specific to neuro-oncology patients. Machine learning models can predict psychiatric outcomes from tumor-induced alterations in functional networks and inflammatory profiles, and AI-driven functional connectivity analysis and predictive biomarker algorithms are revolutionizing diagnostics. Such technologies have the potential of facilitating preemptive intervention in high-risk patients prior to the emergence of severe neuropsychiatric complications [98]. Emerging treatment strategies include neuromodulation therapies. Regarding plasticity reversibility and neural circuits, despite the significant plasticity changes engendered in tumor-induced psychiatric dysfunction such as TMS and DBS, it is still not clear whether plasticity restoration can ultimately restore homeostasis of network functioning. Personalized connectomic mapping has led to the refinement of targeted neuromodulation approaches, a promising frontier in neuropsychiatry [99].

Mechanistic knowledge of neuroinflammation, metabolic dysregulation, and neurotransmitter dysregulation is particularly important for patient management. In patients at high risk of psychiatric deterioration, elevated inflammatory markers, disrupted glutamatergic modulation, or activated kynurenine pathway can serve as warning signs of possible deterioration. Although no study has formally steeped monitoring of any of these biomarkers into clinical practice, making the transition from research to practice may allow less reactive management of symptomatology in conjunction with their oncology therapies whereby they experience less acute crises in addition to less delayed cognitive decline [100].

The field is shifting away from lesion-based models to mechanistic, systems-level frameworks, recognizing that neurological and psychiatric symptoms in brain tumor patients are not incidental comorbidities but rather structural manifestations of tumor biology. Future integrative investigations, merging insights from molecular neuro-oncology, computational neuroscience, and precision psychiatry, will establish the road map for mechanism-driven individualized interventions targeting circuit-level oncological and psychiatric disease burdens concurrently [101]. To summarize the current and emergent methods addressing diagnosis and treatment, Table 2 below offers a selection of modalities and interventions highlighted in the literature. The information serves as a general reference framework, identifying the general principle, current state of clinical readiness, and some key strengths and considerations. It is not intended to be comprehensive, and the relevance of each approach will also depend on the unique aspects of the patients, existing resources, and the ongoing development of the evidence.

## 3. Clinical Presentations: Psychiatric Syndromes and Symptoms

### 3.1. Mood and Anxiety Disorders

Some of the most common psychiatric manifestations in patients with a brain tumor are mood and anxiety disturbances which commonly manifest before neurological deficits and can delay diagnosis. In contrast to primary psychiatric disorders, affective symptoms in this group arise from tumor-specific neurobiological disturbances including neuroimmune activation, metabolic reprogramming, cerebrovascular changes, and large-scale network breakdown [16]. These mechanisms confer novel affective syndromes that do not completely correspond to canonical mood or anxiety disorders, often displaying atypical symptomatology, treatment resistance, and co-occurring cognitive or behavioral changes. The new data support tumor-specific depression and anxiety syndromes with some tumor subtypes, particularly those that demonstrate unique molecular, metabolic, and inflammatory signatures that may trigger syndromes of tumor-specific depression and anxiety and require precision psychiatry approaches for neuro-oncology [102].

Brain tumor patients may develop depression in an insidious manner which are largely independent of psychosocial stressors, making their condition distinct from primary major depressive disorder (MDD) by both clinical trajectory and neurobiological substrate. Also the concept of glioma-related depression (GADep) is new, which proposes that the metabolism of the tumor itself is changing the dynamics of neurotransmitters, particularly in IDH-mutant gliomas where the oncometabolite 2-hydroxyglutaramate (2-HG) accumulates and drives epigenetic silencing of important serotonergic and glutamatergic genes [103]. This creates a cognitive-affective syndrome with anhedonia, psychomotor retardation, and emotional blunting that is frequently refractory to conventional antidepressant treatments [104].

These results underscore the importance of analyzing tumor-specific metabolic alterations as central mediators of neuropsychiatric dysfunction rather than simply to psychosocial distress and/or the emotional burden of malignancy. In IDH-mutant gliomas, the aberrantly dimerized mutant IDH enzyme alters cellular wiring that results in excessive production of 2-hydroxyglutarate from α-ketoglutarate [105]. This pathological accumulation impairs the function of α-ketoglutarate-dependent dioxygenses, including TET enzymes and histone demethylases, resulting in global DNA and histone hypermethylation. Consequently, genes involved in serotonin, glutamate, and synaptic plasticity regulation become epigenetically repressed, causing abnormalities in mood regulation, cognitive processing, and emotional responses in patients [106].

The following figure (Figure 2) summarizes this tumor-driven metabolic and epigenetic reprogramming and indicates how IDH mutations induce neurotransmitter imbalance along with hypermethylation mediated by 2-HG. This framework sets the stage for the development of targeted metabolic and epigenetic therapy to reverse tumor-induced neuropsychiatric dysfunction by elucidating the mechanistic underpinnings of Glioma-Associated Depression.

CNS-related GCC is dysregulated in certain tumors at or outside the CNS (e.g., lung, ovarian, testicular) and results in paraneoplastic mood dysregulation (PMD) with onconeural antibodies targeting neural receptors, observing severe, refractory depression and/or rapid-cycling affective symptoms that are not as yet classified. These cases are frequently mistaken for refractory mood disorders of unclear origin until an oncology workup reveals an underlying malignancy. Detecting CSF biomarkers (e.g., anti-Hu, anti-Ma2, and anti-NMDA receptor antibodies) is vital to distinguishing immune-mediated depression and primary affective disorders to enable immune-targeted interventions before irreversible neural damage set in [107].

Increasing data link gut–brain axis dysregulation to tumor-associated depression and suggest that changes in microbiota profiles, tryptophanto–kynurenine shifts, and systemic inflammation are involved in mood disturbances. Gut dysbiosis caused by brain tumors disrupt microbial production of short-chain fatty acids (SCFAs) and pro-inflammatory cytokines, which in turn affects the homeostasis of glutamate, serotonin, and GABA (through vagus nerve-mediated signaling) [108]. Another effect of this metabolic shift is derived from tryptophan metabolism, because tumors apparently dominate the production of kynurenine rather than serotonin, resulting in the generation of massive amounts of quinolinic acid—a neurotoxic compound that has been associated with suicidality and cognition problems. Meanwhile, these findings emphasize the importance of supplementing the neurotransmitter-oriented therapeutic approaches with microbiome-targeted and anti-inflammatory therapies in the case of depression associated with tumors [109].

This event is also thought to play a role in depressive symptoms as progressions in cerebral blood flow regulation to the tumor impact on neurovascular coupling in brain regions critical to affect. Imaging studies, especially PET, indicate that a number of tumors chronically hypoperfuse and utilize energy inefficiently, especially in the anterior cingulate cortex and dorsolateral prefrontal cortex, correlating with anhedonia, fatigue, and executive dysfunction [110]. The relationship between tumor-related hypometabolism in frontal–limbic circuits and depression may prompt potential therapeutic applications of vascular-modulating agents, including those targeting VEGF, or hyperbaric oxygen therapy, which could ultimately improve mood via neurovascular homeostasis [111].

As a clinical scenario, anxiety disorders in brain tumor patients have special characteristics differentiating them from primary anxiety disorders, commonly being represented as chronic autonomic dysregulation, enhanced fear responses, and increased sensory hypersensitivity. Functional neuroimaging studies have shown that tumors located in the amygdala-prefrontal inhibitory network lead to deficits in top-down modulation of emotional states, which contributes to a state described as amygdala hyperactivity syndrome, characterized by exaggerated startle reactions and exaggerated responses to threat stimuli. This mechanism has been implicated in treatment-resistant anxiety syndromes in glioma and metastatic brain tumor patients, in whom traditional serotonergic and GABAergic interventions may be inadequate [112]. One of the apparent sources of tumor-induced anxiety is excessive glutamatergic pooling in peritumor tissue, notably observed with gliomas, where glutamate in abundance leads to an excitatory environment within the neural interstitium [113]. Unlike primary anxiety disorders, which may be responsive to benzodiazepines and SSRIs, tumor-associated anxiety may be modulated more effectively by NMDA receptor antagonists (e.g., memantine, ketamine) or neurosteroids (e.g., allopregnanolone), which modulate inhibitory–excitatory balance at the synaptic level [114].

Newly diagnosed brainstem and diencephalic tumors are another layer of complexity as they impact autonomic control centers in the locus coeruleus, hypothalamus, and raphe nuclei cause unprovoked panic attacks, tachycardia, and hyperarousal states [115]. They often present with marked physiological symptoms of anxiety but few worry-related cognitions, suggesting a predominance of disturbed brainstem mediation of norepinephrine and serotonin output over mediated anxiety generation at the cortical level. Characterization of such tumor-specific autonomic anxiety syndromes is pivotal for directing proper treatment, given that beta-blockers and alpha-2 adrenergic agonists (e.g., clonidine) may provide more directed approaches to treating underlying neurophysiological dysregulation than traditional psychiatric agents do [116]. An especially underappreciated contributor of anxiety in neuro-oncology patients is corticosteroid-induced neuropsychiatric dysfunction, which can be seen in up to 40% of patients being treated with high-dose dexamethasone for peritumoral edema. Long-term corticosteroid administration hampers HPA axis homeostasis, enhances glutamatergic excitability, and modifies amygdala–prefrontal connectivity, inducing symptoms such as anxiety, restlessness, emotional instability and, rarely, steroid-induced psychosis [117]. These effects may continue even after the end of steroid tapering, indicating that early psychiatric monitoring and the use of adjuncts with mood-stabilizing effects, like lithium or atypical antipsychotics, may be warranted in patients at high risk [118].

The increasing awareness of tumor-associated mood and anxiety syndromes as separate clinical entities has prompted the creation of AI-based predictors that combine information from functional neuroimaging, inflammatory biomarker profiling, and cognitive performance to predict which patients are at greatest risk of severe affective symptoms [119]. Machine learning algorithms are now trained on large-scale neuro-oncology datasets to make predictions about susceptibility to depression or anxiety using tumor location, molecular profile, and metabolic status, paving the way for early, preemptive psychiatric interventions before such symptoms are functionally significant [120]. Likewise, precision psychiatry in neuro-oncology is being pioneered through pharmacogenomic screening and real-time electrophysiological monitoring to selectively optimize antidepressants and anxiolytics for individual patients according to the receptor changes induced by their tumors. These advances are reshaping the treatment landscape away from empirical psychopharmacology to biologically informed personalized therapeutic strategies that target the specific neurochemical, metabolic, and immunological drivers of neuropsychiatric dysfunction in brain tumor patients [121].

The great potentiality of tumor-associated mood and anxiety disorders is identified as being mechanistically different from primary psychiatric mood and anxiety conditions, with significant implications for diagnosis, treatment, and further research. As affective symptoms are frequently discovered prior to the identification of a tumor, the implementation of psychiatric screening into regular neuro-oncological workups may allow earlier tumor diagnosis and potential for improved long-term outcomes [112]. The emphasis on tumor metabolism, neuroimmune activation, and large-scale network dysfunction as fundamental drivers of affective pathology opens entirely new points of medical intervention, ranging from neuroimmune modulators and glutamatergic interventions to microbiome-targeted therapies that stand to drastically reshape the role of psychiatric care within the neuro-oncology practice milieu [16].

### 3.2. Psychotic Symptoms and Delirium

The neuropsychiatric manifestations of primary and metastatic tumors in the brain represent some of the most diagnostically and therapeutically challenging clinical features neuro-oncology, as clinical psychotic symptoms and delirium reflect core disruptions of neural network integrity, neurotransmitter regulation, and systemic homeostasis. While primary psychotic disorders are often attributed to developmental or genetic predisposition, tumor-related psychosis and delirium stem from structural and functional disruptions, including neuroinflammation, metabolic dysregulation, and BBB permeability alterations that are all tumor-location specific [122]. These sequelae are not merely secondary psychiatric manifestations but rather significant symptoms of underlying neurobiological instability that can proceed and predict neurological decompensation, paraneoplastic syndromes, or systemic metabolic failure. Recent advances in functional neuroimaging, computational psychiatry, and immune biomarker profiling are opening new avenues for earlier identification of at-risk patients, distinction between tumor-associated and primary psychotic syndromes, and the development of precision-targeted interventions [123].

In brain tumor patients, psychosis is characterized by hallucinations, delusions, formal thought disorder, and disorganized behavior and is a result of an interruption in dopaminergic, glutamatergic, and limbic-prefrontal connectivity. In contrast to Schizophrenia that develops progressively due to neurodevelopmental aberrances, tumor-associated psychoses can manifest in acute or subacute manners, depending if the underlying mechanism is progressive peritumoral excitotoxicity, network disconnection, or immune mediated dysfunction [124].

Tumors localized in the basal ganglia, nucleus accumbens, or OFC can result in dysregulation of the meso-limbic dopamine system with consequent aberrant dopamine transmission, impairment of salience attribution, and hyperreactivity to emotional stimuli, often causing paranoid ideation, ideas of reference, grandiosity, or persecutory delusions. Unlike primary psychotic disorders, where dopaminergic dysregulation is typically intrinsic and chronic, tumor-associated psychosis frequently fluctuates with the progression of the disease (or peritumoral edema, or corticosteroid use), making its course highly variable [125].

Paraneoplastic schizophrenia-like syndromes (PSS) are an extreme form of tumor-associated psychosis, driven by onconeural antibodies focused on prions and dopamine-related neural circuits. Patients with lung, ovarian, or testicular cancers frequently develop anti-Hu or anti-Ma2 antibody-associated psychotic syndromes, which present with sudden-onset paranoia, agitation, and cognitive impairment and, if untreated, often evolve into catatonia or mutism [15]. These cases do not respond to typical dopamine antagonists but improve dramatically with immuno-modulating therapy, such as rituximab, intravenous immunoglobulin (IVIG), or exchange [125], illustrating the importance of screening CSF antibodies to distinguish immune-mediated psychosis from primary psychiatric disorders [126].

In glial tumors and in metastatic tumors, such as breast carcinoma that infiltrate the hippocampus, anterior cingulate cortex, and medial temporal lobe glutamate, dysregulation has been identified as a major mechanism in tumor-associated psychosis. Glioblastomas over-express the system xc- cystine/glutamate transporter, causing excess glutamate release and chronic excitotoxicity in peritumoral neural circuits, which cause hallucinatory experiences, sensory distortions, and cognitive disorganization [127]. In contrast to dopamine-driven psychotic disorders, hallmark features of glutamate-driven psychosis include perceptual disturbances (often either visual or auditory), dream-like dissociation and hyperperplexity in relation to their environment [128].

The most severe form of glutamatergic dysfunction-associated psychosis is by anti-NMDA receptor encephalitis, which is most often associated with ovarian teratomas or small-cell lung cancer. In this condition, NMDA receptor-specific autoantibodies disrupt glutamatergic neurotransmission, giving rise to a progressive neuropsychiatric syndrome that may start with agitation and paranoia but can progress to severe catatonia and autonomic instability [129]. It is important to distinguish this syndrome from primary schizophrenia as NMDA receptor encephalitis is reversible with immunotherapy, and untreated cases may suffer lethal neurological deterioration [130].

Outside of the risk for neurotransmitter imbalance, emerging evidence highlights the relevance of white matter disconnection syndromes to tumor-associated psychosis, whereby tractography-based analyses show that gliomas are capable of severing critical pathways involved in reality monitoring and emotional processing [131].

The uncinate fasciculus, which connects the OFC and amygdala, is often disrupted in the case of frontal and anterior temporal lobe tumors, leading to delusional misidentification syndromes [132]. Likewise, the superior longitudinal fasciculus, critical for the processing of language and auditory self-monitoring, is associated with thought disorder and auditory hallucinations when disrupted. Resting-state fMRI studies in patients with tumors of the posterior cingulate and precuneus have also shown altered DMN connectivity, correlating with symptoms of paranoia, derealization, and self-referential distortion, in a further confirmation that tumor-related psychosis is not merely a neurotransmitter imbalance but rather a failure of large-scale networks that mediate perception of self and cognitive coherence [133].

Delirium is an acute disturbance of the attention, arousal, and executive function spectrum and is a very common but underappreciated sequelae that may herald systemic metabolic derangement and/or inflammatory overload, visceral organ dysfunction, or ultimately, neurological collapse among the brain tumor population. In contrast to psychosis, which reflects discrete cortical dysfunction, delirium demonstrates a global breakdown in neural synchrony and systemic homeostasis, requiring urgent medical assessment [134]. Substantial metabolic derangements (e.g., hypercalcemia, hyponatremia, and dysregulated glucose metabolism) are common provokers of acute confusional states in patients with metastatic cancer. Endocrine dysfunction also contributes, specifically hypercortisolemia with its associated agitation and steroid psychosis from ectopic ACTH secretion, as well as pituitary adenomas causing hypothyroidism or adrenal insouciance, which can result in profound executive dysfunction and hypoactive delirium [135].

BBB disruption in gliomas and metastatic tumors permits peripheral cytokines, au-toantibodies, and systemic toxins to penetrate the CNS, which fosters global neuroinflammatory priming. Our results support the hypothesized mechanism of inflammation mediating delirium, with higher levels of IL-1β, IL-6 and TNF-α being significantly related to higher levels of delirium severity—supporting the possible use of anti-inflammatory agents as novel therapies to target this potentially reversible condition. Due to growing awareness of the mechanistic differences between tumor-associated psychosis and delirium versus primary psychiatric diagnoses, AI-enabled diagnostic models have been created to predict an individual patient’s neuropsychiatric decline using neuroimaging, inflammatory biomarkers, and metabolic parameters [136]. These models allow for the early identification of patients at risk for psychosis or delirium and therefore provide an opportunity for targeted preemptive interventions prior to the complete onset of symptoms. Furthermore, network-level approaches such as neuromodulation with repetitive transcranial magnetic stimulation (rTMS) and DBS have been theorized for treatment of refractory tumor-associated psychosis and delirium, especially in patients unresponsive to traditional pharmacotherapy [137]. Shifts in pharmacological approaches are also under way toward glutamate-modulating agents (perampanel, a selective AMPA receptor inhibitor, and riluzole, a metabotropic glutamate receptor modulator) which could offer targeted relief from tumor-driven hallucinations and paranoia [138,139].

Identification of tumor-related psychosis and delirium should lead to urgent neuroimaging and metabolic evaluations. In patients with a high risk of rapid deterioration—such as patients with temporal lobe lesions, tumors with multifocal disease, or those with rapid changes in neurological status—CSF antibody testing may be valuable as there may be some evidence that indicates it could exclude autoimmune encephalitis. Involving psychiatry and neuro-oncology early can help with safe antipsychotic use, seizure prophylaxis, and monitoring during tumor-directed treatment [140].

That tumor-associated psychosis and delirium will be recognized as fundamental signals of underlying neuroinflammatory and network perturbation, rather than solitary neurotransmitter imbalance, will require a paradigm shift in neuro-oncology toward mechanistically driven, precision-based psychiatry. As AI-based neuroimaging and molecular profiling continue to improve, integrating early detection of disease-modifying processes, druggable immune alteration, and network-based neuronal shaping will transform the way in which neuropsychiatric symptoms are employed as markers of psychiatric care in brain tumor patients, making sure that these types of neuropsychiatric symptoms will be treated as core pathology and not just a secondary phenomenon [141].

### 3.3. Cognitive and Behavioral Changes

Cognitive and behavioral deficits in patients with brain tumors remain some of the most complex and diagnostically difficult manifestations of neuro-oncological disease. Cognitive and behavioral syndromes differ from focal neurological deficits, which are due to local structural disruption, as they result from systems failure, inflammatory modulation of synaptic function, metabolic reprogramming, and the disconnection of large-scale white matter [142]. These patients may show cognitive alterations well in advance of overt neurological signs and may serve as an early biomarker of tumor presence. As we gain insights into these syndromes through advances in functional neuroimaging, computational neuroscience, and precision medicine, we are learning that tumor-related cognitive and behavioral dysfunction are not merely a consequence of mass effect but rather a disease process itself [143].

The frontal lobes are responsible for planning goal-directed behavior, inhibiting impulses, solving problems, motivating us to action, modifying social appropriateness. Tumors involving this area interfere with these higher-order processes in subtle or profoundly disabling ways, depending on the site and extent of involvement [144]. Recent developments in tractography-based white matter mapping and network neuroscience demonstrate that frontal dysfunction in patients with brain tumors is referable not just to direct gray matter impairment but also to the progressive dissolution of cortico-subcortical connectivity. The OFC is an important center for emotion regulation, risk assessment, and social behavior [145]. Tumors in this area generate a characteristic syndrome characterized by behavioral disinhibition, impulsivity, inappropriate social behavior, and emotional instability. In contrast to primary psychiatric disorders, which are characterized by emotional dysregulation associated with introspective distress, orbitofrontal dysfunction results in a diminished awareness of one’s behavioral change that manifests in reckless decision-making, compulsivity, and environmental hyper-reactivity [146].

Recent functional MRI studies indicate that orbitofrontal disruption affects connectivity between the amygdala and prefrontal regulatory circuits, resulting in unfiltered emotional expression, lack of risk evaluation, and a heightened response to both reward and punishment. This modified reward processing can lead to hypersexuality, financial irresponsibility, compulsive gambling, or extreme sensation-seeking behaviors, paralleling certain facets of frontotemporal dementia, but with different tumor-dependent pathophysiology [147].

The DLPFC represents a central hub for executive function, working memory, problem-solving, and abstract reasoning. Tumors that encroach on this region induce a syndrome of cognitive inflexibility, attentional deficits, and severe challenges adjusting to changing situations. In contrast to primary neurodegenerative disorders, which result in a gradual onset of executive dysfunction, DLPFC dysfunction associated with tumor progression may have an abrupt onset, especially due to peritumoral edema or metabolic changes impacting frontoparietal network efficiency [148]. Evidence from cognitive neuroscience studies indicates that tumors in the DLPFC decrease frontoparietal control network’s ability to synchronize, and are associated with impairments in task switching, strategy formation, and updating working memory. PET imaging has shown that tumors located within this region can result in regional hypometabolism contributing to the disruption of dopamine-dependent executive circuits, explaining the common inadequacy of stimulant-based cognitive enhancers in restoring function in these patients. The attention paid to dopamine-modulating interventions, specifically targeting tumor-induced executive dysfunction, is a new area of research and potential treatment [149].

Tumors involving the medial frontal cortex and anterior cingulate cortex (ACC) lead to a syndrome characterized by debilitating apathy, abulia (loss of willpower), and psychomotor slowing [150]. It is often misdiagnosed as depression, but unlike primary mood disorders, in which sadness and distress are prominent, medial frontal dysfunction results in an alarming absence of internal motivation, spontaneous thought generation, and emotional engagement [151]. Task-independent functional connectivity is disrupted between medial frontal regions and the meso-limbic reward system by tumors within this area, causing a deficit in dopamine-mediated motivation. Although dopamine agonists and DBS aiming at the anterior cingulate-striatal loop may prove potential treatment options in the future for tumor-induced apathy, relatively little has been investigated in this domain in the literature [152].

Episodic memory encoding, emotional modulation, and personality stability are regulated functionally by the temporal lobes, which are vulnerable to cognitive and affective alterations from tumor invasion. Unlike frontal lobe dysfunction that mainly affects executive processing, temporal lobe tumors cause significant distortions in memory recall, emotional responsiveness, and self-identity integration [153]. Tumors involving the mesial temporal structures, most commonly gliomas or metastases, that compromise the hippocampus and parahippocampal gyrus cause anterograde amnesia, spatial disorientation, and confabulatory memory distortions. In contrast to primary amnestic disorders like Alzheimer’s where memory impairment is progressive and diffuse, tumor-induced hippocampal dysfunction typically has an asymmetrical presentation, depending on whether the dominant or non-dominant hemisphere is involved [154,155].

New findings from hippocampal-subcortical connectivity studies suggest tumor-specific disruptions at the level of theta-band oscillatory activity may drive retrieval deficits, representing a tractable target for future network-based neuromodulation strategies [156,157].

The amygdala and anterior temporal cortex play key roles in the processing of fear, the detection of emotional salience, and social cognition. Tumors located here often lead to emotional dysregulation, increased anxiety responses, and have been shown to alter personalities. Unlike psychiatric specific mood disorders, where emotional lability is often context dependent, temporal lobe disfunction leads to a more persistent basal shift in emotional tone with exaggerated fearful response, emotional lability, or, paradoxically, flattened emotional response [158]. DTI studies indicate that tumor-induced disconnection of the amygdala from the prefrontal cortex results in deficits in top-down emotion regulation and creates a neurological predisposition to overreact to slights and misperceive social dynamics. This mechanistic understanding of tumor-driven emotional dysregulation opens the door to precision neuromodulatory therapies that might restore functional connectivity between the amygdala and regulatory circuits [159].

Applied to functional neuroimaging, metabolic profiling, and inflammatory marker analysis, machine learning algorithms are being used to predict cognitive decline trajectories in brain tumor patients long before reaching a clinical endpoint of deterioration. These methodologies enable foundational intervention schedules, individualized cognitive rehabilitation, and specially developed pharmacological strategies [160].

Preliminary data indicates that chronic neuroinflammation may play a role in cognitive impairment in neuro-oncology practice. Other investigational agents directed at microglial over-activation (minocycline, IL-6 inhibitors) and astrocytic glutamate deregulation (riluzole, AMPA receptor antagonists) also appear promising for the prevention of TCI [161], listed as follows:Repetitive rTMS of the dorsolateral prefrontal cortex is being investigated for executive dysfunction in frontal gliomas [162].Anterior cingulate cortex DBS appears promising in the treatment of tumor-induced apathy and motivation deficits [163].

Cognitive and behavioral dysfunction in patients with brain tumors is not just a secondary result of mass effect but, rather, a dynamic, progressive disease involving metabolic, immune, and network-level derangements [122]. The marriage of sophisticated neuroimaging, machine learning-based diagnostic analytics, and the growing repertoire of precision neuromodulation strategies will enable precise interventions that will place tumor-associated cognitive and behavioral syndromes at the forefront of oncological care, rather than as an afterthought. As patient-focused care begins to receive as much attention as patient-centric research, and as more psychiatric/neuropsychological expertise is built on top of our existing technical foundations, this new angle will have wide-ranging implications for how neuro-oncology adopts cognitive neuroscience, computational modeling of cognition, and therapeutic neuromodulation-based interception [164].

### 3.4. Subtle Personality Changes

In practice, subtle changes in personality may be the first and most diagnostically precarious manifestation of a brain tumor, emerging years before deficits of cognition, overt behavioral dysregulation, or deficits in neurological function. These changes can be insidious enough that they become dismissed as stress-, age-, and personality-related manifestations, when in fact they are often reflective of profound neurobiological alterations engendered by tumor-induced derangements in neural connectivity, neurochemical homeostasis, and systemic metabolic and immune regulation [122]. Whereas personality alterations in psychiatric diseases are often episodic and reactive in nature, tumor-related personality changes are chronic in nature, are typically progressive in nature, and are characteristically resistant to conventional psychiatric treatments, making their recognition crucial for early diagnosis and treatment [165].

Though an early description of personality changes attributed them to mass effect and direct cortical invasion, increasing evidence strongly suggests that such disturbances result from multifactorial large-scale brain networks, neurotransmitter systems, white matter integrity, and neuroinflammatory cascades. Connectomics, high-resolution neuroimaging, artificial-intelligence (AI)-driven behavioral analytics, and metabolic neuropsychiatry neoplasm-associated behavior and personality changes can lead to the pathologic derangement of self-regulation, social cognition, and emotional balance, contributing considerably to the fundamental alteration of personality even with a small, slow-growing tumor [166]. These findings have important implications for the way we should think about tumor-associated personality changes, as not just by-products of neurological disease, but as primary mechanistic changes in brain function that could serve as early markers of tumor activity [167].

What is critical for personality regulation is the dynamic interplay of the prefrontal cortex, limbic system, basal ganglia, and respective white matter pathways, each utilizing self-regulation, emotional stability, and impulse control as an element of social behavior. Tumors involving these areas (either directly or through long-range disconnection effects) lead to progressive distortions to personality traits, causing patients to become much more irritable, less socially motivated, more emotionally volatile, or less self-aware and empathic [168]. Dysfunction is not just a function of tumor size and location, but also of its overall influence on distributed brain networks and its potential to trigger second-order neurobiological processes such as excitotoxicity, neuroimmune activation, and metabolic reconfiguration [169].

Brain personality changes in brain tumor patients are tightly related to white matter disruption, while studies using DTI and lesion network mapping show much less of a correlation with gray matter damage. Tear pads of three large white matter tracts involved in the structural disruption of personality dysfunction have been detected in these studies in tumor patients. Disruption of the uncinate fasciculus, a structure responsible for connecting the OFC with the amygdala and limbic system, can lead to emotional instability, under-regulated response inhibition, and impaired control over impulsivity [170]. Damage of the cingulum bundle connecting the medial prefrontal region to the anterior cingulate and areas of limbic structures results in amotivation, anhedonia, and extreme lack of self-activation. Neurodegeneration of the inferior fronto-occipital fasciculus, an essential pathway that integrates executive control and reward processing, leads to dysregulated risk assessment, poor social judgment, and increased impulsivity [171].

There have been few studies on the relationship between peritumoral white matter hyperintensities (WMH) and personality change, which have traditionally been analyzed or regarded as imaging findings of either a predictable or incidental nature, such as peritumoral edema. They are indicative of subclinical neuroinflammatory mediators as measures of chronic neuroinflammation and glial dysfunctions or impaired neuronal repair, and, as a potential biomarker, they also have the ability to predict personality decline in the absence of structural injury on conventional imaging [172]. In neuro-oncology, speculative new approaches include AI-based volumetric assessments of WMH, which may facilitate earlier therapeutic interventions for patients at higher risk of personality disturbances [173].

Chronic inflammation of the nervous system has recently gained traction as a critical instigator of tumor-associated alterations in personality, as accumulating data point at chronic microglial activation, dysregulation of cytokines, and neuroimmune crosstalk as key disruptors of personality homeostasis. Translocator protein (TSPO) ligands for positron emission tomography (PET) imaging have demonstrated glial pathology initiated by gliomas and metastases by demonstrating widespread microglial activation in the OFC, mPFC, and ACC, all of which are critically important to personality regulation [174]. Increased concentrations of IL-6, TNF-α, and IL-1β have been demonstrated to affect serotonergic and dopaminergic transmission, impair synaptic plasticity, and change emotional processing, resulting in increased fighting, impulsivity, and social withdrawal [175]. These results implicate tumor-induced neuroinflammation as a possible therapeutic target for stabilizing personality function. New therapeutics targets that are based on the contribution from systemic inflammation to personality disorder of neutrophils can be devoted to the purposes of IL-6 inhibitors and TNF-α blockade [176].

The gut–brain axis is being increasingly acknowledged as an influential mediator of personality regulation, and recent observations support the notion that brain tumors can drive changes in microbiota composition that affect neurotransmitter homeostasis via peripheral immune signaling. Recent studies demonstrate that glioblastoma patients present gut microbiota diversity alterations with decreased SCFA-producing species and increased pro-inflammatory species, which is correlating with increased aggression, increased social withdrawal, and decreased emotional flexibility. These findings indicate that microbiome-targeted interventions, such as FMT and probiotic-based therapies, may have clinical potential as adjunctive treatments for the management of tumor-induced personality changes [177]. Moreover, brain tumors modify tryptophan metabolism, diverting the pathway towards kynurenine production and away from serotonin synthesis, resulting in elevated levels of neurotoxic metabolite, quinolinic acid. This finding also supports the idea that metabolic approaches may ameliorate the behavioral changes (tumor personality) seen in malignant transformation due to elevated quinolinic acid levels in prefrontal–limbic circuits leading to increased impulsivity, emotional inhibition, and likelihood of reactive aggression [178].

AI and machine learning in personality tracking are transformative in pinpointing subtle changes in behavior in patients with neuro-oncology. AI can provide speech analysis of an individual (particularly their patterns of vocal intonation) as well as tracking the movements of their facial expressions and the statistics from biosensors measuring their physiological responses, that can detect changes in the function of their personality even before they have clinically manifested. These technologies could be used to predict personality deterioration trajectories, enabling preemptive neuroprotective interventions [179]. One promising intervention is neuromodulation strategies targeting large scale networks of personality. High-definition transcranial direct current stimulation (HD-tDCS), targeted at the medial prefrontal cortex, is entering clinical trials as an experimental therapy to augment behavioral self-regulation while DBS around the anterior cingulate cortex is being investigated for reversing tumor-acquired apathy and impulsivity. These interventions represent the next wave of precision psychiatry for neuro-oncology patients [180].

The brains of patients with tumors are also affected by the tumor, and subtle changes in personality in such patients are not just psychological reactions to having an illness, but the natural results of tumor-driven network dysfunction, immune dysregulation, and metabolic shifts. The synergy of AI-based behavioral tracking, gut–brain modulation, neuroimmune-targeted therapeutics, and neuro-modulation approaches is the future of early detection and personalized intervention. By focusing on preemptive neuro-oncological surveillance rather than reactive management, these advances can impact clinical out-comes, leading to the recognition of personality disturbances as an essential part of brain tumor pathology and not an incidental finding [181].To consolidate the diverse clinical presentations and their underlying mechanisms, we aimed to present a synthesis table (Table 3) mapping each major psychiatric syndrome to its proposed pathophysiological basis, diagnostic clues, and potential management approaches.

## 4. Diagnostic Challenges and Management

### 4.1. Diagnostic Challenges

Validation of tumor-associated neuropsychiatric syndromes as distinct clinical entities is a fundamental challenge within neuro-oncology and neuropsychiatry, warranting a paradigm shift away from traditional, symptom-based psychiatric nosology toward biological, precision-based diagnostic systems. In contrast to primary psychiatric conditions driven by inherent neurochemical defects, developmental disorders, or psychosocial environmental factors, tumor-associated psychiatric symptoms result from progressive structural, functional, and metabolic disturbances in the brain [182]. These disruptions take the forms of widespread network dysconnectivity, peritumoral excitotoxicity, neuroinflammatory signaling, and metabolic shifts, all of which together drive psychiatric syndromes that may resemble major depressive disorder, bipolar disorder, schizophrenia spectrum disorders, and neurocognitive disorders [183]. Yet, though they share phenotypic features with primary psychiatric disorders, tumor-induced neuropsychiatric syndromes have unique mechanistic underpinnings, surprising trajectories, and variable responsiveness to conventional psychotropic medications. Not recognizing these differences may result in a prolonged misdiagnosis, an unnecessary experience of inappropriate psychiatric interventions, and substantial delays in oncology treatment, which profoundly impact patient outcomes [184].

The challenge of recognizing tumor-mediated psychiatric symptoms is compounded by the occurrence of psychiatric manifestations that may precede overt neurological deficits; many patients initially seek care through psychiatric services, rather than neurology or oncology. The fact that these patients frequently undergo prolonged psychiatric management before neuroimaging subsequently demonstrates an intracranial neoplasm. Particularly with respect to cases involving frontal, temporal, or subcortical regions, emerging evidence suggests that psychiatric symptoms may represent some of the first markers that can be observed following the onset of brain tumors [185]. Because of this, there is an urgent need to implement standardized screening protocols in order to identify tumor-driven psychiatric presentations early in their course. Measures tailored to facilitate the distinction of disorders wherein psychiatric symptoms arise from physical causes will only become possible with more recently developed technologies, such as lesion-network mapping, functional neuroimaging, biomarker analysis through liquid biopsy, and AI-enabled diagnostic algorithms, all of which have allowed for unprecedented precision and early detection in diagnosis of structurally driven neuropsychiatric syndromes when compared to primary psychiatric illnesses [186].

Distinguishing tumor-associated neuropsychiatric syndromes from primary psychiatric disorders is one of the most diagnostically challenging tasks in neuropsychiatry. Their nature differs substantially from that of the classic psychiatric disorders, which tend to have episodic, stress-modulated or developmental trajectories and partially or fully remit with appropriate pharmacological or psychotherapeutic interventions [16]. In contrast, tumor-associated psychiatric symptoms often herald an insidious onset, progressive worsening, and resistance to standard psychiatric treatment, mirroring the biologic evolution of the tumor itself. In contrast to primary psychiatric illnesses, which are usually episodic in nature and fluctuate in severity in response to environmental and psychosocial triggers, tumor-driven psychiatric symptoms escalate over time in a manner directly associated with progression of the tumor mass, peritumoral inflammation, and metabolic dysregulation [187].

Another vital diagnostic marker can be seen from the neurocognitive profile of patients who develop tumor-induced psychiatric symptoms. Although primary mood and psychotic disorders tend to maintain core cognitive abilities except in times of acute exacerbation, tumor-associated psychiatric disorders often involve insidious yet progressive deficits in executive function, attentional control, and working memory. Such deficits are often out of proportion to the severity of mood or psychotic symptoms, pointing to an underlying structural etiology [188]. Neuropsychological evaluations often demonstrate cognitive inflexibility, psychomotor retardation, and deficits in set-shifting and decision-making, which are mediated by fronto-striatal and limbic-prefrontal circuits frequently compromised in patients with gliomas and metastatic brain tumors. All patients presenting with deteriorating problem-solving abilities, goal-directed behavior, or social cognition that is unexplained, particularly in the absence of a clear psychiatric history, should undergo urgent evaluation for possible structural lesion [189].

Tumor-associated psychiatric syndromes also differ from primary psychiatric illnesses in that they are associated with atypical sensory-perceptual experiences. In contrast to hallucinations occurring with primary psychotic disorders, which often have well-organized thematic narratives, tumor-related hallucinations are often persistent, multimodal, and associated with transient cognitive confusion, micrographia, or visuospatial disturbances [190]. Patients with temporo-parietal tumors commonly report episodes of déjà vu, jamais vu, or episodic ictal fear, the latter features that are very suggestive of an underlying epileptiform process rather than primary schizophrenia spectrum disorder. The new development of visual or olfactory hallucinations in an older person without a previous psychotic history should always raise concern for an organic etiology—an urgent neuroimaging workup is warranted in this case [191].

Over the years, the distinction between tumor-associated psychiatric symptoms and primary psychiatric disorders has evolved with the advent of neuroimaging which can now reveal network-level dysfunctions that can extend beyond the primary tumor lesion site. Standard MRI and CT remain the gold standard for identifying mass lesions; however, emerging techniques such as functional MRI (fMRI), PET, DTI, and lesion-network mapping (LNM) provide invaluable insights into the neurobiological mechanisms underlying tumor-driven neuropsychiatric syndromes [192].

Asymmetrical lesions likely disrupt behavior by the more conventional mechanism of mass effect and tissue loss but may also do so via direct disruptions of these systems, as shown through lesion-network mapping which has transformed psychiatric neurosurgery by showing the disruption of large-scale functional systems in the brain by tumors. In contrast to conventional lesion-symptom models, which assume a one-to-one relationship between lesion location and function, LNM reveals that small tumors can elicit dramatic psychiatric symptoms by disconnection of major hubs within the DMN, limbic system, and fronto-parietal control networks [193]. Recent studies revealed that frontotemporal gliomas disrupt fronto-striatal and limbic-thalamic connectivity resulting in executive dysfunction, apathy, and impaired goal-directed behavior; all of these aspects are mistakenly diagnosed as major depressive disorder or fronto-temporal dementia [194]. Tumors of the temporal lobe also impair default mode—salience network coupling, leading to deficits in social cognition, paranoia, and emotional instability that resembles schizophrenia spectrum disorders. Subcortical tumors on the basis ganglia and thalamus can lead to neuropsychiatric syndrome of psychomotor slowing, emotional flattening, and altered decision-making, closely resembling subcortical depression or parkinsonian apathy [195].

PET imaging, especially fluorodeoxyglucose (FDG)-PET, builds on this, delineating regional metabolic and functional failure in critical neuropsychiatric circuits. In contrast to primary psychiatric disorders, which tend to have limited kinetic deficits, tumor-associated psychiatric syndromes associate with focal metabolic hypoactivity in the OFC, anterior cingulate cortex, and meso-limbic structures that recapitulates the projections disturbed by lesion-network mapping [196]. PET imaging should also be considered in this context and could be integrated into MRI-based assessments of structural and functional brain properties in order to achieve a comprehensive approach to the detection of tumor-driven neuropsychiatric dysfunction, as this appears to markedly improve diagnostic accuracy [197].

Recent evidence suggests that EVs (including exosomes) that are secreted from tumor cells may be useful as early biomarkers of neuropsychiatric sequelae in patients with brain tumors. Freed gliomas, metastases, and even systemic malignancies exocytose exosomes filled with microRNAs, inflammatory cytokines and metabolic end products that faithfully alter neurotransmission, synaptic plasticity, and neuroinflammatory pathways [198]. Exosomes from glioblastoma containing miR-21/155 have also been found to inhibit the expression of synaptic plasticity genes in the PFC, resulting in executive function and mood dysregulation. Metastatic tumors release exosomes that carry tau- and amyloid-associated proteins, enhancing neurodegenerative-like alterations to the limbic system [199]. Transfer of IL-1β and TNF-α from tumor cells to astrocytes and microglia via exosomes promotes persistent neuroimmune activation, which has been directly associated with depressive and anxiety-like behavior. Assessment of tumor-derived exosomes in plasma and CSF by liquid biopsy techniques may facilitate non-invasive recognition of tumor-associated psychiatric disorders prior to structural changes on neuroimaging [200].

Considering the multifaceted nature of tumor-related psychiatric syndromes, precise identification of these entities necessitates collaboration between neuro-oncologists, psychiatrists, neurosurgeons, neuroradiologists and neuropsychologists. Integration of advanced neuroimaging, lesion-network mapping, liquid biopsy biomarkers, and AI-based diagnostic tools are the next horizon of precision psychiatry in neuro-oncology [201]. Dedicated neuro-oncology psychiatric screening programs will need to be established to identify patients early, and to provide timely intervention to improve patient outcomes. A potential next step could be that a neurologist helps cancer patients with a psycho-neuronal approach, a move from a symptom-driven diagnostic paradigm to a biological, precision paradigm, which could lead to a new era in neuro-oncology psychiatry with improved diagnostic and therapeutic precision [202].

### 4.2. Management Strategies

It is however worth briefly commenting on the management of tumor-associated neuropsychiatric symptoms, which remains one of the most complex and rapidly evolving terrains in neuro-oncology and neuropsychiatry, calling for a precision-based, multimodal strategy that takes into account the interplay of neuroanatomical disruption, neurotransmitter dysregulation, systemic inflammation, metabolic perturbations, and oncological treatment effects [203]. In contrast to the functional origin of primary psychiatric disorders, tumor-induced psychiatric disorders are caused by structural network disintegration, excitotoxicity, immune-mediated synaptic remodeling and metabolic changes, and palliative strategies must transcend the conventional psychopharmacological paradigm [204].

In the past, the psychiatric manifestations of patients with brain tumors were treated empirically and psychotropic regimens were a heterogeneous non-specific approach of an attempt to control the symptoms rather than addressing the underlying pathology. However, recent research indicates that tumor-induced neuropsychiatric syndromes possess specific receptor pharmacodynamics, neurotransmitter imbalances, and circuit-specific dysregulations that necessitate mechanism-driven therapies [205]. The journey has led to an evolution in the future of psychiatric care in neuro-oncology, including advances in precision psychopharmacology, neuromodulation, neuroimmune-targeted therapies, AI-driven behavioral monitoring, and metabolic optimization. The aim of management is not simply to palliate psychiatric symptoms, but also to stabilize neural function, suppress further cognitive decline, improve quality-of-life, and facilitate oncological treatment adherence [206]. The management of psychiatric symptoms in neuro-oncological patients through pharmacological means remains unique, as tumor progression, peritumoral neuroinflammation, and increased BBB permeability can impact pharmacological responses and drug dosages, especially with reliance on the same drugs used for cancer treatment. Standard psychiatric pharmacology—which assumes stable receptor profiles and neurotransmitter function—is often inadequate or poorly tolerated in neuro-oncology patients, requiring neurobiologically-informed, receptor-specific, circuit-targeted psychotropic selection [207].

These recent findings have shown that tumor-induced psychiatric symptoms are not simply secondary emotional responses to disease but are in fact directly modulated by disturbances in localized neurotransmitter circuits. Tumor-induced apathy, cognitive rigidity, and anhedonia may result from dopaminergic hypofunction in meso-cortical pathways, leading to the hypothesis that dopaminergic agonists (e.g., pramipexole or ropinirole) may be more efficacious than SSRIs in such patients [208]. Likewise, tumors affecting orbitofrontal and amygdaloid circuits interfere with serotoninergic modulations, precipitating emotional lability and aggression, thus leading directions towards administration of 5-HT1A partial agonists (buspirone, for example) as a possible replacement of classical SSRIs [209]. Glutamatergic excess and peritumoral excitotoxicity, particularly in gliomas, have been linked to neuropsychiatric instability, with newer data supporting the use of NMDA receptor modulators such as memantine to allow the stabilization of cognitive and affective function within the context of neurocognitive rehabilitation efforts [127].

Pharmacogenomic studies have demonstrated abnormalities in the metabolism of psychotropic drugs in patients with brain tumors, adding complexity to treatment selection. There are polymorphisms of CYP450 enzymes that can alter the metabolism of antidepressants, anxiolytics, and antipsychotics, which means that the dose of these classes needs to be adjusted and that agents should be selected on the basis of genetic screening for metabolic capacity. As a solution, pharmacogenomic modeling is now made possible with the advent of AI, allowing for the prediction of optimal drug regimens according to tumor profiles, receptor densities, and metabolic variations for maximum efficacy with minimal detrimental effects [210]. Neuroimmune dysregulation has emerged as a main driver of psychiatric dysfunction in patients with brain tumor, and chronic neuroinflammation alters neurotransmitter function, disrupts neural plasticity, and induces persistent mood and cognitive deficits. Tumor-driven immune activation-inducing cytokine-mediated synaptic remodeling, microglial activation, and astrocytic dysfunction are major pathophysiological mechanisms linking tumors in the CNS with neuropsychiatric dysfunction. Recent trials have examined whether anti-inflammatory and immuno-modulatory agents may reduce tumor-associated psychiatric symptoms, with promising results [211].

In fact, TNF-α inhibitors including etanercept and infliximab have been found to have antidepressant and cognitive-sparing effects in various neuroinflammation-based depressive syndrome and cognitive impairment in cancer patients, and their effect appears to result from a manner that focuses on the patho-physiology of depression rather than simply treating the symptoms. Tocilizumab-mediated IL-6 blockade has also been explored as a countermeasure for immune-mediated neuropsychiatric dysfunction, particularly in patients treated with ICIs where cytokine-induced encephalopathy exacerbates psychiatric features [212]. In glioma models, the microglial inhibitor minocycline has shown neuroprotective effects, reducing overactive microglial activation and synaptic degradation, and thus acts as an adjunct treatment for cognitive preservation [213]. Given that tumor-driven neuroinflammation alters dopamine transporter function, attenuates serotoninergic transmission, and potentiates glutamate excitotoxicity, the inclusion of neuroimmune-targeted therapies to the standard of neuro-oncology psychiatric care is a paradigm shift in treatment philosophy, heralding a transition from purely symptom suppression to mechanistically principled intervention at the neurobiological level [214]. Neuromodulation techniques, in addition to pharmacological and immunological strategies, have emerged as helpful tools to restore the disrupted neural networks of tumor patients. Unlike drug-based approaches that can result in systemic side effects, neuromodulation enables regionally restricted, circuit-specific therapy to counteract tumor-induced network dysregulation [12].

Repetitive rTMS of the dorsolateral prefrontal cortex is effective in alleviating depression, cognitive flexibility, and working memory deficits among glioma patients, and the mentioned variables showed the most significant improvement in glioma patients with left frontal damage. Theta-burst TMS of the anterior cingulate cortex represents a novel therapeutic strategy for tumor-associated apathy and executive dysfunction, with the potential to partially reverse impaired fronto-limbic connectivity and increased motivational drive [162]. In a similar vein, DBS has been studied as a novel intervention for disabling and treatment resistant neuropsychiatric symptoms due to brain tumors, specifically orbitofrontal dysfunction, of which nucleus accumbens DBS has demonstrated effectiveness restoring goal-directed behavior. The task has also been piloted in patients who underwent targeted stimulation of the subthalamic nucleus for the modulation of impulsivity and aggression in tumor-induced fronto-subcortical disinhibition [215]. Machine learning behavioral analysis via natural language processing and facial biometrics can reveal subtle changes in mood, cognition, and personality long before clinical symptoms develop, enabling preemptive intervention rather than treatment after the fact. Data scientists are already applying machine learning algorithms to multimodal neuroimaging, EEG data, and digital phenotyping to correlate with and predict which tumor patients are at the highest risk of severe psychiatric complications, paving the way for targeted early interventions. Pharmaco-genomics driven by AI is becoming the next step of optimization, enabling drug regimens to be tailored to individual metabolic and receptor profiles of the patient, thus ensuring ideal therapeutic responses with lesser toxicity and potential drug-to-drug interactions [179]. This is a progression towards the next frontier of progress in precision psychiatry in neuro-oncology as it transitions from a static, symptom reduction model to a fluid, classic data-driven model that evolves with the patient’s changes in neurochemical state [216].

## 5. Prognosis, Outcomes, and Future Directions

Traditionally, the prognosis of human brain tumor patients has been evaluated based on oncological parameters such as tumor grade, histopathological classification, and response to standard treatments. But increasingly, evidence is mounting that neuropsychiatric symptoms exert independent effects on functional outcomes, treatment adherence, and survival [217]. Psychiatric dysfunction associated with tumors is no longer considered a secondary effect but a primary contributor to quality of life, treatment-resistance, and long-term outcome trajectories. The paradigm of integrated neuropsychiatric care, predictive modeling of cognitive and affective outcomes, and targeted the neurotherapeutic strategies evolved due to recognition of psychiatric stability as a prognostic biomarker [218]. Studies have shown that patients with untreated depression, cognitive impairment, or psychosis have lower adherence to oncological therapies, higher incidence of post-surgical complications, and lower survival rates. The mechanisms have broad implications for therapies for patients with cancer through neuroimmune dysregulation, HPA-axis dysfunction, tumor-derived metabolic reprogramming, and disorganization of reward-motivation circuits that affect resilience to therapy. The need for continuous psychiatric supervision and intervention at every phase of therapy is now recognized as an essential pillar of neuro-oncological care [108].

As AI-guided prognostic models, biomarker-based psychiatric risk classification and connectome-based precision psychiatry have all made headway in their respective fields, a gradual emergence of personalized therapeutic strategies targeting both psychiatric and oncological disease trajectories is on the horizon. The future of neuro-oncology psychiatry lies in its integration with predictive analytics, targeted neuromodulation, and network-based psychiatric interventions, ensuring that neuropsychiatric stability dominate the standard oncological endpoint rather than remaining an ancillary concern [53,219].

### 5.1. Prognosis and Quality of Life: The Impact of Psychiatric Symptoms on Survival and Treatment Adherence

The psycho-neurological–tumor axis has long been disregarded, but more recent studies have demonstrated psychiatric disturbance as an independent measure of negative oncological prognosis [220,221]. It has been shown that depression and anxiety disorders negatively impact compliance with chemotherapy, radiotherapy, and rehabilitation programs resulting in poor oncological treatment responses and higher early mortality rate [222]. Neuroimmune activation mediates cancer progression in a connected pathway whereby psychiatric dysfunction causes pro-tumorigenic immune activation. Chronic stress and unresolved depression induce pro-inflammatory signaling, leading to increased circulation of IL-6, TNF-α and IFN-γ, further driving malignancy, angiogenesis, and immune evasion [223]. Increased glucocorticoid resistance in immune cells induced by HPA-axis dysregulation mediated feedback between stress and absence of effective anti-tumor immunity enhancing this vicious cycle. This demonstrates that stabilization of psychiatric function measures not only a quality-of-life marker but may also provide a favorable milieu for immunotherapies resulting in improved overall survival [224]. Functional neuroimaging studies performing fMRI and PET metabolic profiling have further demonstrated that patients with refractory depression demonstrate diminished metabolic firing in anterior cingulate and orbitofrontal systems mediating motivated cognition, influencing meso-limbic reward circuitry, collectively predicting proclivity toward fatigue, anhedonia, and motivational inertia inhibiting therapeutic engagement. Analyzing neuroimaging features using machine learning approaches can now allow the identification of the patients at highest risk of psychiatric decompensation with tumor progression, facilitating early interventions prior to the symptoms developing into disabling conditions [225].

In the field of psychiatry, personalized risk stratification models integrating neuroimaging, immune profiling, and AI-driven behavioral analytics are being developed to recognize high-risk patients, tailor psychiatric treatment choice, and enhance functional recovery over time. Such predictive models are the new frontier of neuro-oncology psychiatry to assure that psychiatric optimization is an integral part of precision cancer care [226].

### 5.2. Outcomes over Time: The Evolution of Psychiatric Symptoms Across the Tumor Trajectory

The evolution of psychiatric symptoms among patients with brain tumors is highly dynamic and relies on a number of interacting variables from tumor progression, treatment modality, neuroplasticity, and systemic metabolic changes. In contrast to primary psychiatric illnesses that can be chronically persistent or recurrent episodic, tumor-associated neuropsychiatric dysfunction evolves along a disease-modulated time course, adapting to temporal changes in the oncological and neurobiological landscapes [188].

Pre-diagnostic psychiatric symptoms—including changes in personality, apathy, or subtle cognitive deficit—preceded neurological signs by months to years and thus are candidates for an early marker in brain tumors. In early disease stages, psychiatric symptoms predominantly arise from mild network dysregulation and peritumoral excitotoxicity, while in advanced stages, tumor-induced neuroinflammation, metabolic stress, and diaschisis effects underpin progressive deterioration in mood, cognition, and behavior [203]. The psychiatric sequelae of surgery are typically diverse, depending not only on the decamping of the tumor site but also on the extent of resection and what compensatory mechanisms of neuroplasticity are activated. Tumor removal resolves symptoms for some patients, especially those where mass effect was the primary cause of psychiatric dysfunction [227]. However, in other instances, surgical resection produces new or exacerbates existing psychiatric symptoms due to further disruption of neural networks, white matter tract damage, or postoperative neuroinflammation. It is now applied in predicting post-surgical neuropsychiatric risk and guiding neurosurgical strategies that are aplastic with respect to psychiatric morbidity [228]. A subset of patients develops neurocognitive and affective disturbances months or years after radiotherapy and chemotherapy, attributed to radiation-induced microvascular injury, glial dysfunction, and impaired neurogenesis. Survivorship studies have shown that in brain tumor patients, long-term psychiatric morbidity is especially high, and ongoing psychiatric follow-up is required long after oncological remission has been achieved [229].

Large-scale, AI-powered, longitudinal neuropsychiatric tracking is enabling researchers and clinicians to plot the accurate temporal evolution of psychiatric symptoms in neuro-oncology. Combining data from real-time behavioral monitoring, multimodal neuroimaging, and inflammatory biomarker analysis, these predictive models may allow psychiatric deterioration trajectories to be predicted, and preemptive interventions initiated that are tailored to the individual patient’s neurobiological risk profile [201].

### 5.3. Future Directions and Research Gaps: The Next Frontiers in Neuro-Oncology Psychiatry

Several key research gaps persist despite growing recognition of the need to maintain neuropsychiatric stability in patients with brain tumors. The lack of a standardized framework for assessing and managing psychiatric symptoms in neuro-oncology underscores the need for large, prospective studies. Future clinical trials should assess the most suitable psychotropic, neuromodulatory, and immune-targeted therapies tailored to optimize psychiatric function in this group [230]. Advances in imaging biomarkers, such as submillimetric functional imaging, neuroinflammatory PET tracers, and lesion-network mapping, will both contribute to advance the modulation of early psychiatric dysfunction sequelae in patients with tumors. Such models will also allow for precision psychiatry approaches that personalize psychiatric intervention based on a patient’s specific neurobiological profile, e.g., AI-driven diagnostic models and biomarker-based systems of risk stratification [231].

The integration of neuro-oncology and psychiatry within a single clinical framework is one of the urgent needs of the field. Neuro-oncology psychiatric services should be credentialed, as well as incorporated into standard of care so that psychiatric assessment and intervention would be integrated into oncological workflows: from diagnosis to long-term survivorship. This includes lobbying for increased funding for neuropsychiatric research, the establishment of AI-vised predictive analytics, and the initiation of integrated care prototypes that fuse neuro-oncological with computational neurobiological, immuno-psychiatry, and neuromodulation [232].

As a result, and through a multidisciplinary, data-driven, and mechanistically informed approach, neuro-oncology psychiatry has the potential to evolve from a historically ignored area, into a forward-leading, precision-driven specialty that maximizes both psychiatric and oncological outcomes [233].

The literature regarding tumor-related psychiatric syndromes is hampered by heterogeneity in study populations, diagnostic criteria, and methods of symptom assessment. Most cohorts tend to be small and single center, meaning that generalizability is limited. Psychiatric endpoints are predominately classified as secondary endpoints, which leads to subsequent incomplete or inconsistent characterization of symptom trajectories [234]. A variety of causal–mechanistic associations, such as cytokine signatures, disruptions from connectome analyses, or exosomal biomarkers, are primarily supported by correlational data with unknown sequence or directionality. Models based in artificial intelligence remain in early stages of development that will require refinement, as there is a risk of outcome bias stemming from variations in imaging protocols, variations in preprocessing, and variations in the demographic representation of imaging datasets [235]. Finally, when studies are designed to be interventional, few incorporate prospective designs using psychiatric outcomes as the primary endpoint with the potential for direct translation into an integrated neuro-oncology psychiatry approach. To improve the knowledge gaps highlighted throughout this review, both the assessment of neuro-oncological psychiatric outcomes should use standardized assessments, while also being multicentral and using prospective clinical trials to embed neuro-oncological psychiatry measures with the oncological outcomes [236].

## 6. Conclusions

### 6.1. Summary of Current Evidence

In the last 20 years, research has fundamentally changed our understanding of the relationship of psychiatric dysfunction to a neoplasm. Cancer diagnosis (malignant disease) may be seen as a secondary phenomenon (brought on by the psychological reaction to the malignant diagnosis), whereas psychiatric dysfunction has been increasingly viewed as a direct neurobiological outcome of the presence or progression of the tumor. Utilizing a wide variety of tumor types, the empirical direction is reasonably consistent between possible direct consequences of large-scale network disruption, alterations in neurotransmitter systems, neuroimmune dysregulation, and metabolic reprogramming as major contributors to neuropsychiatric syndromes. More importantly, these neurobiological features are not only important for patient comfort or quality of life; they fundamentally affect core oncologic processes—treatment compliance, neurological hardiness, complication rate, and ultimately overall survival.

### 6.2. Diagnostic Advances and Transformation

An interesting, most likely underappreciated, finding is that psychiatric symptoms often precede diagnostic imaging detection of tumors, which challenges the existing framework that emphasizes focal neurological detection as a greater good. Subtle changes in mood, executive control, personality, and social cognition, particularly limited to arrangements affecting the frontal lobe, temporal lobe, or limbic area, may surely reflect the earliest clinical manifestations of tumor biology as well. If the new functioning psychiatric disorders go unrecognized, they may experience a protracted period of poor psychiatric treatment prior to imaging detection of the structural lesion. The next-generation possibilities utilizing lesion network mapping (circuitry network), AI-based imaging (structural and functional), liquid biopsy biomarkers (including tumor-derived extracellular vesicles), and metabolic imaging offer the possibility of demonstrating a tumor-related psychiatric syndrome several months in advance of a standard structural scanning and detecting a biological impact. 

### 6.3. Prognostic Importance and Mechanisms

The prognostic importance of psychiatric symptoms in neuro-oncology is now well established. Untreated depression, cognitive dysfunction, and neurobehavioral dysfunction are independent prognostic factors for lower chemotherapy adherence, higher surgical complications, higher rates of attrition with radiation, and poorer overall survival. Mechanistic work has shown that neuroimmune dysregulation, dysregulation of the hypothalamic–pituitary–adrenal (HPA) axis, metabolic dysregulation, and breakdown of reward-motivation circuitry conspire to reduce physiologic and psychological resilience to oncologic therapies. This process demonstrates that psychiatric stability is not a supportive adjunct, but rather a contributory determinant of therapeutic benefit.

### 6.4. Therapeutic Possibilities and Dynamic Nature of Presentation

Tumor-associated psychiatric dysfunction is dynamic, varying together with changes in tumor biology, treatment-related effects, and neuroplastic changes. Soon after diagnosis, psychiatric symptoms will often develop slowly from diffuse network dysfunction as opposed to focal mass effect. As disease continues to develop, aggressiveness of neuroinflammatory cascades, excitotoxicity, and neural trauma induced by treatment can escalate symptoms or lead to a different psychiatric phenotype. Some patients will achieve full psychopathological recovery following resection of the tumor, while others will develop new or persistent symptoms as a result of surgical disconnection, radiation-associated gliopathy, and neurotoxicity as a result of chemotherapy.

We are entering a new frontier in neuro-oncology psychiatry. New tools will allow for biomarker-based risk stratification, machine learning-based prognostication, precision psychopharmacology, and targeted neuromodulation for unprecedented anticipatory treatments that will adapt to the patient’s ever-evolving neurobiological state. These personalized treatments have the potential to improve immune resilience, increase neuroplasticity, and optimize patient outcomes in both the psychiatric and oncological domains.

### 6.5. Research Gaps and Opportunities

Despite the advancements made during the past 10 years, there remain observable research gaps and opportunities. There is an urgent need for multicenter prospective cohort studies to incorporate high-resolution neuroimaging, immune signature profiling, tumor-derived exosome analysis, and continuous compliance, behavioral, and cognitive monitoring. Future studies should treat psychiatric outcomes as primary endpoints, not just studies of reducing symptoms, but also in the context of mechanistically targeted neuromodulation, neuroinflammatory therapies, and precision pharmacology that track neurophysiology in real-time. When it comes to advancing the discipline, standardization is essential—to achieve a level of consistency, it will be important for us to have internationally transformative diagnostic family criteria or classifications, to develop diagnostic stages and models, and to provide a framework for supporting and characterizing care for neuro-oncology psychiatry intervention will be a means of minimizing differences in care based on clinical context.

### 6.6. Vision for the Discipline

The next decade provides an opportunity for the full integration of neuro-oncology psychiatry into the broad thematic frameworks of computational neuroscience, the emerging field of neuro-immunology and precision oncology. With advancements in connectomics, ways to modify inflammatory pathways, and predictive AI behavioral analytics, we are entering a phase in which psychiatric care can transition from a reactive form of the management of symptoms, to a proactive form of psychiatric surveillance—identifying dysfunction in the preclinical phase when it is possible to intervene in patients for the best outcomes. The ultimate aim will be for this kind of model of psychiatric care to not only be patient specific, and a mechanism based and enabled by technology, but to be embedded both from a diagnosis phase through to survivorship.

In this future prospect, psychiatric stability will be considered to be equally, if not more important than, then as a secondary comfort measure, a primary oncological outcome in and of itself, in that it will not only be important for achieving tumor control, but also for longer term survival. Neuro-oncology psychiatry will be a fully integrated collaborative medical specialty, with early detection, targeted intervention and ongoing longitudinal monitoring as a standard of care, for every patient to ensure that AIM incorporates outcome-optimal neuropsychiatric therapy.

## Figures and Tables

**Figure 1 ijms-26-08114-f001:**
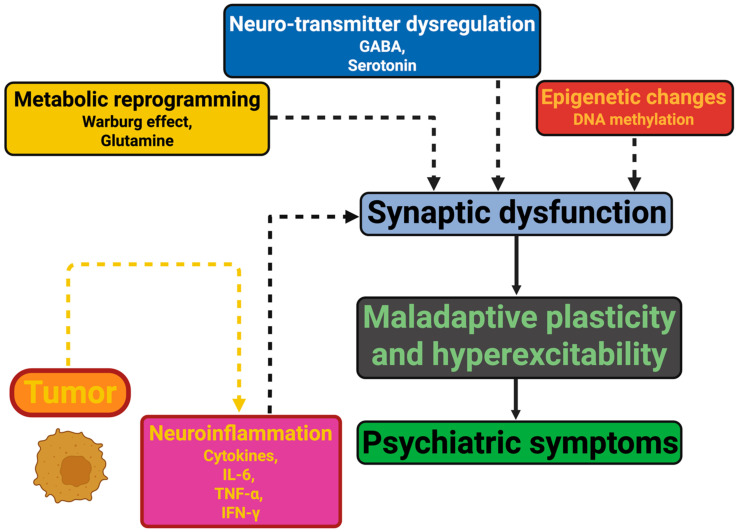
Pathophysiological mechanisms linking brain tumors to psychiatric symptoms.

**Figure 2 ijms-26-08114-f002:**
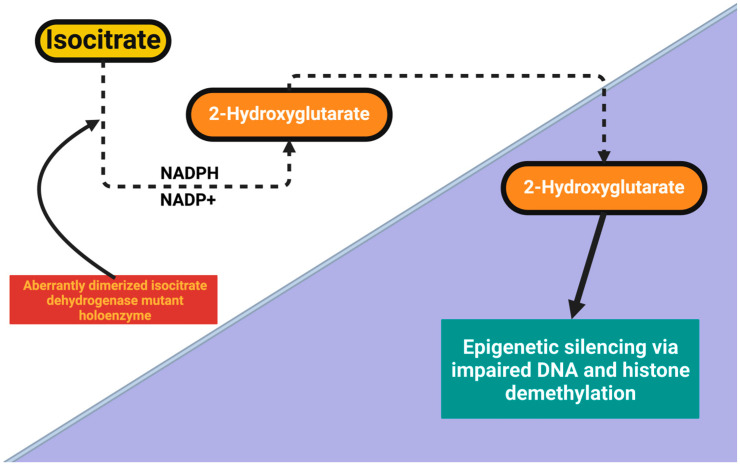
This figure depicts the disrupted metabolic pathway caused by mutant IDH enzyme activity, emphasizing its cascading effects on epigenetic regulation and neurotransmitter balance. Under normal physiological conditions, isocitrate is converted into α-ketoglutarate (α-KG), a vital cofactor for DNA and histone demethylation enzymes. However, in IDH-mutant gliomas, the enzyme undergoes abnormal dimerization, resulting in excessive production of 2-hydroxyglutarate (2-HG), an oncometabolite that profoundly alters gene expression. As 2-HG accumulates, it functions as a competitive inhibitor of α-KG-dependent dioxygenases, such as TET DNA demethylases and Jumonji-domain histone demethylases. This inhibition promotes global DNA and histone hypermethylation, locking tumor cells into a state of transcriptional repression. The silencing of genes involved in serotonergic, glutamatergic, and neuroplasticity-related pathways contributes to the mood disturbances, cognitive rigidity, and emotional flattening characteristic of Glioma-Associated Depression (GADep).

**Table 1 ijms-26-08114-t001:** A synthesis of high-impact original research and clinical trials investigating the neuropsychiatric manifestations of various brain tumors. It categorizes findings based on tumor type, associated psychiatric syndromes, pathophysiological underpinnings, and key clinical implications, providing a structured, evidence-based overview of how distinct tumor characteristics influence neuropsychiatric function. The table purpose is to highlight mechanistic insights derived from neuroimaging, molecular profiling, immune markers, and neurotransmitter alterations, offering a translational perspective on the interplay between tumor biology and psychiatric symptomatology. By incorporating prospective cohort studies, randomized controlled trials, and AI-driven predictive models, this table underscores the role of emerging diagnostic techniques and targeted therapeutic interventions in refining psychiatric management in neuro-oncology. It also emphasizes how lesion localization, neuroimmune dysregulation, metabolic reprogramming, and epigenetic modifications contribute to symptom heterogeneity, paving the way for precision psychiatry approaches tailored to the neurobiological landscape of each tumor subtype.

Tumor Type	Psychiatric Manifestations	Pathophysiological Insights	Key Findings and Clinical Implications	References
Glioblastoma (GBM)	Depression, executive dysfunction, emotional blunting	Glioma-induced glutamate excess disrupts fronto-striatal circuits	High glutamate levels in peritumoral regions correlate with depressive symptoms; NMDA receptor antagonists (memantine) show promise in symptom reduction.	[66,67,68]
Low-Grade Gliomas	Personality changes, apathy, cognitive rigidity	Disruption of fronto-limbic pathways and anterior cingulate hypometabolism	Functional MRI (fMRI) reveals decreased ACC activation; dopaminergic modulation with pramipexole improves motivation and goal-directed behavior.	[69,70,71]
Meningioma (Frontal and Orbitofrontal)	Disinhibition, impulsivity, hypersexuality, compulsive behaviors	Orbitofrontal cortex compression leads to impaired impulse control and social cognition	High-resolution tractography demonstrates disconnection of the uncinate fasciculus in symptomatic patients; neuromodulation (TMS) targeting the DLPFC improves inhibitory control.	[72,73,74,75]
Pituitary Macroadenomas	Psychosis, mood instability, cognitive slowing	Endocrine dysfunction (hyperprolactinemia, hypercortisolism) induces dopaminergic dysregulation	Dopamine agonists (cabergoline) reverse psychotic symptoms in prolactinomas, but cognitive impairments persist in Cushing’s patients’ post-surgery.	[76,77,78,79]
Brain Metastases (Lung, Breast, Melanoma)	Anxiety, panic attacks, depressive symptoms, cognitive deficits	Multifocal disruption of limbic circuits, BBB permeability alterations, paraneoplastic autoantibodies	Higher systemic IL-6 and TNF-α levels correlate with severe affective disturbances; checkpoint inhibitors exacerbate neuropsychiatric dysfunction in 20% of cases.	[80,81,82]
Medulloblastoma (Pediatric Tumors)	Executive dysfunction, emotional dysregulation, ADHD-like symptoms	Cerebellar-prefrontal network disruption impairs cognitive flexibility	Post-treatment ADHD-like symptoms linked to cerebellar dysfunction; targeted cognitive rehabilitation improves attention and executive function.	[83,84]
Thalamic Gliomas	Apathy, psychomotor slowing, altered consciousness states	Thalamic relay dysfunction impairs arousal and motivation circuits	PET imaging demonstrates severe metabolic hypoactivity in anterior thalamus correlating with anhedonia; stimulant-based interventions (modafinil) improve alertness.	[85,86,87]
Glioblastoma (IDH-Wildtype vs. IDH-Mutant)	Cognitive impairment, depression, emotional blunting	IDH-mutant gliomas exhibit preserved neurocognitive function compared to IDH-wildtype	Epigenetic changes in IDH-mutant gliomas lead to reduced psychiatric burden; DNA methylation therapies show promise for cognitive preservation.	[88,89,90]
Anti-NMDA Receptor Paraneoplastic Encephalitis (Ovarian Teratomas, Small-Cell Lung Cancer)	Severe psychosis, aggression, catatonia, autonomic instability	Autoimmune attack on NMDA receptors disrupts excitatory neurotransmission	Plasma exchange and immunotherapy (rituximab) rapidly resolve psychiatric symptoms; early diagnosis via CSF biomarkers is crucial.	[91,92,93]
Multiple Tumor Types	Predictive modeling of psychiatric deterioration	Machine learning applied to multimodal imaging, speech, and inflammatory markers	AI model predicts psychiatric symptom emergence 3–6 months before clinical manifestation with 85% accuracy; potential for preemptive psychiatric intervention.	[94,95,96]

**Table 2 ijms-26-08114-t002:** Synthesis of selected diagnostic modalities and therapeutic interventions relevant to the condition under review. Each entry highlights the main target or operational principle, the present level of clinical readiness, and practical considerations that may influence use. The table is intended to complement the narrative text by offering a side-by-side view of diverse approaches without implying preference or guideline status.

Modality/Intervention	Primary Target or Principle	Clinical Readiness	Advantages	Limitations/Considerations
MR Venography (MRV)	Visualization of venous sinus patency and flow	Widely available	Non-invasive, high spatial resolution	May miss dynamic changes; contrast use in some protocols
Intracranial Pressure Monitoring	Continuous measurement of ICP waveform and amplitude	Routine in select neurosurgical settings	Direct physiological data	Invasive, risk of infection
Diffusion Tensor Imaging (DTI)	Assessment of white matter integrity and AQP4-related changes	Research to early clinical adoption	Detects microstructural changes	Requires advanced analysis; interpretation variability
CSF Cytokine Panel	Measurement of inflammatory mediators	Research use	Provides immunological insight	Limited availability; unclear clinical thresholds
Venous Sinus Stenting	Restoration of venous outflow	Specialized centers	Rapid ICP reduction in selected cases	Procedural risk; not universally applicable
CRISPR-Mediated AQP4 Repair	Gene editing to restore perivascular localization	Preclinical	Targeted, potentially disease-modifying	Long-term safety unknown; ethical considerations
Aquaporin Modulators	Pharmacologic modulation of AQP4 activity	Early clinical/experimental	Non-invasive, adaptable	Off-target effects; variable patient response
Anti-inflammatory Biologics	Modulation of microglial and cytokine activity	Limited clinical evidence	Addresses inflammatory component	Cost, immuno-suppression risk

**Table 3 ijms-26-08114-t003:** Summary of key pathophysiological domains potentially contributing to the observed clinical spectrum, with associated molecular mechanisms, illustrative diagnostic approaches, and examples of emerging or established therapeutic strategies. This synthesis is intended as a broad conceptual framework drawn from the current literature, acknowledging that many aspects remain under investigation and that applicability may vary across individual cases and contexts.

Pathophysiological Node	Molecular Mechanisms	Diagnostic Signatures	Therapeutic Strategies	Evidence Level
Venous outflow impairment	Jugular/venous sinus congestion, ↑ central venous pressure	MRV, jugular Doppler, ICP waveform	Endovascular stenting, CSF shunting	Clinical series
AQP4 depolarization	Loss of perivascular localization, altered astrocytic endfeet	Immunohistochemistry, DTI	CRISPR-mediated repair, gene therapy	Preclinical
Glymphatic slowdown	Reduced CSF–ISF exchange, protein aggregation	Contrast MRI, PET tracers	Sleep optimization, aquaporin modulators	Mixed
Neuroinflammation	Microglial activation, cytokine release	PET (TSPO), CSF cytokine panel	Anti-inflammatory biologics	Early clinical
Global access gap	Limited imaging, surgical resources	Health system mapping	Low-cost diagnostics, task-shifting	Policy/implementation

## Data Availability

The data presented in this study are available on request from the corresponding author.

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
