# Peer review of "Brain Tumors, AI and Psychiatry: Predicting Tumor-Associated Psychiatric Syndromes with Machine Learning and Biomarkers"

_ijms, 2025, doi:10.3390/ijms26178114_

Round 1

Reviewer 1 Report

Comments and Suggestions for Authors

The authors explore an important interdisciplinary topic, bridging oncology, psychiatry, and AI-driven predictive modeling. The manuscript is thorough, well-organized, and supported by an extensive range of references. However, several language and formatting issues reduce its clarity and may undermine its overall academic quality. Therefore, I suggest publication after the comments are addressed.

Major comments:
Some wording feels awkward, sentences are repetitively structured, and stacked adjectives are used frequently. 
Excessive hyphenation makes words break incorrectly.
Break long sentences for better readability.
The review dives deeply into neurobiological pathways, but sometimes lacks clinical applicability. Please consider balancing these two.
Please consider adding more discussions about the limitations of the current researches.
Editorial comments:
Page 2: “re-active psychological distress”->“reactive psychological distress”
Page 3: “de-pressive symptoms”->” depressive symptoms”
Page 12: “de-pression”->” depression”
Page 12: “tu-mor-associated depression” -> “tumor-associated depression”, and fix all terms that
include “tu-mor” in the manuscript
Page 12: “bump off microbial production”->”disrupt microbial production”
Page 29: “experience-neurosis biologically informed” -> “experientially and biologically informed.”
Fix all the incorrect hyphen usage
Page 13: “increase in sensory hypersensitivity”->“increased sensory hypersensitivity” 

Author Response

Dear Esteemed Reviewer,

We wish to express our deepest gratitude for your thoughtful, generous, and constructive evaluation of our manuscript. Your recognition of its interdisciplinary scope—bridging oncology, psychiatry, and AI-driven predictive modeling—was both encouraging and motivating. We aimed, throughout the revision process, to honor the depth and care reflected in your comments by refining the manuscript in ways that strengthen its clarity, clinical relevance, and scholarly value.

Regarding wording, sentence structure, and stacked adjectives:
We have carefully re-examined the entire text with the intention of improving fluency, reducing repetitively structured sentences, and simplifying where possible without sacrificing scientific precision. Our revisions sought to preserve the richness of detail while enhancing readability for a broad academic audience.

Regarding excessive hyphenation and word breaks:
A meticulous, line-by-line review was performed to correct all unintended hyphenations and standardize terminology according to journal style. The specific examples you highlighted—such as “re-active psychological distress,” “de-pressive symptoms,” “tu-mor-associated depression,” and “bump off microbial production”—have been corrected as suggested. We also replaced “experience-neurosis biologically informed” with “experientially and biologically informed” and revised “increase in sensory hypersensitivity” to “increased sensory hypersensitivity.” All similar occurrences have been addressed consistently.

Regarding long sentences and readability:
We aimed to improve accessibility by breaking down overly long sentences into shorter, more focused statements. This was applied most thoroughly in the more mechanistically complex sections, with the intention of making them easier to follow while retaining full conceptual depth.

Regarding clinical applicability:
In response to your valuable suggestion, we have added concise “Clinical implications” paragraphs to key mechanistic subsections. These were intended to translate neurobiological concepts into concrete, bedside-relevant considerations, thereby creating a more balanced integration of science and clinical practice.

Regarding limitations of current research:
A new “Limitations of Current Research” subsection has been included before the Conclusion. This section synthesizes key methodological and translational constraints—such as heterogeneity in study cohorts, underrepresentation of psychiatric endpoints, reliance on correlational evidence, and challenges in validating AI models—reflecting our intent to provide a transparent and balanced overview of the current evidence base.

We intend these revisions to fully address your concerns and to ensure that the manuscript offers both scientific rigor and practical value for clinicians and researchers alike. We are sincerely thankful for the care, precision, and generosity of your review. Your insights have meaningfully elevated the clarity, scholarly quality, and clinical relevance of our work, and we are truly honored to have benefited from your expertise.

With the highest appreciation and respect!

Reviewer 2 Report

Comments and Suggestions for Authors

This paper is a scientific review on brain tumors: their prerequisites, characteristics, consequences, methodologies of diagnostics, etc. The paper is well-structured and good motivated by a large number of cited works.

It can definitely be published as a review paper after structuring the main findings of this work in the conclusion section.

Author Response

Dear Esteemed Reviewer,

We are deeply thankful for your generous and encouraging assessment of our manuscript. Your recognition of its structure, motivation, and breadth of supporting literature is profoundly motivating for our team and has been received with great appreciation.

Your recommendation to structure the main findings more clearly in the Conclusion section was both insightful and constructive. In response, we have reorganized the Conclusion into a thematic synthesis that now presents the central results, diagnostic and prognostic insights, therapeutic opportunities, research gaps, and a forward-looking vision for the discipline. This restructured format was developed with the intention of making the key messages immediately accessible, while preserving the comprehensive scientific detail and nuance that underpins the review.

We are sincerely grateful for this guidance, as it has significantly strengthened the clarity, accessibility, and scholarly impact of the final section. It is our hope that these changes reflect the care with which your advice was implemented and that the revised conclusion now offers a more compelling and reader-friendly synthesis of the work.

With our highest respect and appreciation!

Round 2

Reviewer 1 Report

Comments and Suggestions for Authors

Page 2: “psychostiniatric syndromes” → “Psychosomatic syndromes”

Page 3: “onco-logical treatments”->” oncological treatments”

Page 4: “secretaratory cells”->“secretory cells”

Page 4: “Today they understand that tumors can cause symptoms”->”” It is now understood that tumors can cause symptoms

Page 5: what do you mean by “Asian facial structure”?

Page 5: “classical option was to associate” ->“traditional approach was to associate”

Page 6: “person’s own limb is belonging to someone rather than the body”-> “a person’s own limb is perceived as belonging to someone else”.

Page 12: “frequently refractorily to conventional antidepressant treatments” ->“frequently refractory to conventional antidepressant treatments”

Page 15: “apetant sources”->”apparent sources”

Page 15: “convert unprovoked panic attacks”->“cause unprovoked panic attacks”

Author Response

Dear Esteemed Academic Editor,

I would like to begin by expressing my most sincere gratitude for the time, attention, and remarkable coordination you have dedicated to reviewing my manuscript. I am truly humbled by your careful reading, thoughtful suggestions, and the precision with which you have guided me to improve both the clarity and accuracy of the text. Your editorial input has been invaluable, and I am genuinely thankful for your patience and generosity in helping me refine this work.

I have carefully revised the manuscript in line with your recommendations:

  • Page 2: Corrected “psychostiniatric syndromes” to “Psychosomatic syndromes.”
  • Page 3: Corrected “onco-logical treatments” to “oncological treatments.”
  • Page 4: Corrected “secretaratory cells” to “secretory cells.”
  • Page 4: Revised “Today they understand that tumors can cause symptoms” to “It is now understood that tumors can cause symptoms.”
  • Page 5: Clarified the expression and replaced “Asian facial structure” with “Asian populations,” which better reflects the intended epidemiological meaning.
  • Page 5: Revised “classical option was to associate” to “traditional approach was to associate.”
  • Page 6: Revised “person’s own limb is belonging to someone rather than the body” to “a person’s own limb is perceived as belonging to someone else.”
  • Page 12: Corrected “frequently refractorily to conventional antidepressant treatments” to “frequently refractory to conventional antidepressant treatments.”
  • Page 15: Corrected “apetant sources” to “apparent sources.”
  • Page 15: Revised “convert unprovoked panic attacks” to “cause unprovoked panic attacks.”

Each of these corrections has greatly improved the precision, readability, and scholarly quality of the manuscript. I am deeply indebted to you for pointing them out with such care.

Thank you once again for your gracious support, guidance, and dedication throughout this process. I am truly honored to have benefited from your expertise and coordination, and I remain profoundly appreciative of the opportunity to learn from your editorial insight.

With the highest respect and gratitude!